# Simultaneous application of enzyme and thermodynamic constraints to metabolic models using an updated Python implementation of GECKO

Jorge Carrasco Muriel,[1,2] Christopher Long,[3] Nikolaus Sonnenschein[1]

**ABSTRACT** Genome-scale metabolic (GEM) models are knowledge bases of the reactions and metabolites of a particular organism. These GEM models allow for the simulation of the metabolism, for example, calculating growth and production yields—based on the stoichiometry, reaction directionality, and uptake rates of the metabolic network. Over the years, several extensions have been added to take into account other actors in metabolism, going beyond pure stoichiometry. One such extension is enzyme-constrained models, which enable the integration of proteomics data into GEM models containing the necessary $k_{cat}$ values for their enzymes. Given its relatively recent formulation, there are still challenges in standardization and data reconciliation between the model and the experimental measurements. In this work, we present geckopy 3.0 (genome-scale model with enzyme constraints, using Kinetics and Omics in Python), an actualization from scratch of the previous Python implementation of the same name. This update tackles the aforementioned challenges, to reach maturity in enzyme-constrained modeling. With the new geckopy, proteins are typed in the Systems Biology Markup Language (SBML) document, taking advantage of the SBML Groups extension, in compliance with community standards. In addition, a suite of relaxation algorithms—in the form of linear and mixed-integer linear programming problems—has been added to facilitate the reconciliation of raw proteomics data with the metabolic model. Several functionalities to integrate experimental data were implemented, including an interface layer with pytfa for the usage of thermodynamics and metabolomics constraints. Finally, the relaxation algorithms were benchmarked against public proteomics data sets in *Escherichia coli* for different conditions, revealing targets for improving the enzyme-constrained model and/or the proteomics pipeline.

**IMPORTANCE** The metabolism of biological cells is an intricate network of reactions that interconvert chemical compounds, gathering energy, and using that energy to grow. The static analysis of these metabolic networks can be turned into a computational model that can efficiently output the distribution of fluxes in the network. With the inclusion of enzymes in the network, we can also interpret the role and concentrations of the metabolic proteins. However, the models and the experimental data often clash, resulting in a network that cannot grow. Here, we tackle this situation with a suite of relaxation algorithms in a package called geckopy. Geckopy also integrates with other software to allow for adding thermodynamic and metabolomic constraints. In addition, to ensure that enzyme-constrained models follow the community standards, a format for the proteins is postulated. We hope that the package and algorithms presented here will be useful for the constraint-based modeling community.

**KEYWORDS** constrained-based modeling, proteomics, thermodynamics

Address correspondence to Jorge Carrasco Muriel, jcamu@dtu.dk.

C.L. and N.S. are employees of Ginkgo Bioworks.

See the funding table on p. 14.

*[This article was published on 16 October 2023 with information missing in the Acknowledgments section. The missing information was added in the current version, posted on 7 November 2023.]*

For those who have taken courses in Biochemistry, Metabolism, and metabolic regulation as undergraduates, the calculation of yields of ATP from a substrate may sound quite familiar. They may remember the effort of memorizing the balance of the different routes in the central metabolism, how these metabolic routes branch out, and the difficulty of clarifying the destiny of the flux of carbon, as it is distributed across the network.

The complexity and ubiquity of these calculations in Biochemistry curricula was noted by M. R. Watson in the 1980s (1). He devised how this process, which he regarded as laborious, could be handled by a computer, both easing the understanding of Metabolism and demonstrating the application of computation in Biochemistry.

Thus, reactions can be rewritten as computations. Consider, for instance, equation (1). In this equation, we have two reactions and four metabolites $\{A, B, C, D\}$.

$$A + 2B \rightarrow C \qquad (1)$$
$$A + C \rightarrow 4D \qquad (2)$$

We can rewrite the reactions as a matrix S using the stoichiometric coefficients (equation (3)).

$$\begin{bmatrix} -1 & -2 & 1 & 0 \\ -1 & 0 & -1 & 4 \end{bmatrix} \qquad (3)$$

Now, by enforcing the steady-state assumption $\mathbf{Sv} = 0$ over the vector of reaction fluxes $\mathbf{v}$, we can convert the metabolic network into a system of linear equations. Further constraints about the reaction directionality and exchange rates can be imposed, turning it into a linear programming (LP) problem (2). By optimizing the so-called biomass function (3), that is, an artificial sink reaction added to the metabolism network that turns essential precursor metabolites into biomass based on empirically determined proportions, growth can be simulated. When the reactions and metabolites of a whole species are gathered into one of these models, we call it a genome-scale metabolic (GEM) model. GEMs are stored in the Systems Biology Markup Language (SBML) (4), which is an open standard for the Systems Biology community.

Far from being mere learning tools in the classrooms, genome-scale metabolic models have proven to be useful in a variety of ways, from strain design for biochemical production, drug-target prediction (5), and improving the understanding of human diseases and the metabolism of different organisms (6).

In addition, GEM models can be used as a platform to integrate omics data. With the use of gene-protein-reaction rules existing in SBML documents, proteins can be associated with their corresponding reactions. Several formulations have been developed to account for proteins in the GEM model (7–11). One of these formulations is GECKO (genome-scale model with enzyme constraints, using Kinetics and Omics): if turnover numbers ($k_{cat}$) are available, GEM models can be extended by including them in the form of inverse stoichiometric coefficients and enzyme concentrations as additional pseudometabolites in the reactions (12, 13) (Eq 4).

$$A + 2B + \frac{1}{k_{cat1}}P_1 \rightarrow C$$
$$A + C + \frac{1}{k_{cat2}}P_2 \rightarrow 4E \qquad (4)$$

Hence, this formulation allows for the inclusion of experimental data in the form of absolute proteomics. This is done by constraining the maximum value of each protein variable to the concentration in the data set. Thus, the solution of fluxes has to be consistent (less or equal) with the maximum saturation of the enzyme working at the maximum rate. Alternatively, if such data are not available, the sum of all protein variables can be constrained to be less or equal to the total amount of protein in the

cell. The latter approach would be equivalent to MOMENT 7 and requires the retrieval of molecular weights of the enzymes in the model since the total protein is usually described as the mass of the cell dry weight ($g/g_{DW}$).

More formally, enzyme-constrained models expand the stoichiometric matrix S by adding new protein "metabolites" $P$. Furthermore, exchange pseudoreactions are added to supply the proteins into the network. The supply is constrained to an upper bound, which may be used to indicate an experimental measurement of the given protein. This results in a linear programming problem of the same time complexity as the original flux balance analysis (FBA), with a smaller solution space.

In addition, a formulation to include thermodynamics called thermodynamics flux analysis (TFA) has also been proposed and implemented in previous publications (14, 15). With the inclusion of reaction Gibbs free energies, TFA constrains the reactions to follow the second law of thermodynamics. Conveniently, metabolomics concentrations can be integrated into the computation of Gibbs free energies.

As mentioned above, GEM models have been successfully stored and disseminated as SBML documents, which state the reactions, metabolites, constraints, and annotations associated with them. Nonetheless, the typing of enzymes in the SBML document remains a challenge. Until now, these enzymes have been stored as *SBML Species*, relying on naming conventions to differentiate them from metabolites. This is error-prone and also hinders their usage within existing COnstraints-Based Reconstruction and Analysis (COBRA) software (16–18). Moreover, the integration of absolute proteomics in enzyme-constrained models is not trivial; it usually yields models that are infeasible for the given reported growth and exchange fluxes, requiring some relaxation of the experimental constraints (19).

In this work, we present software for the incorporation of omics constraints into genome-scale metabolic models with enzymatic parameters: geckopy 3.0. This package—which takes over the limited geckopy Python layer built in 12—tackles the aforementioned problems, providing (i) an SBML-compliant formulation of enzyme pseudometabolites; (ii) relaxation algorithms for reconciling the model with the experimental data; and (iii) an integration layer with existing academic software—pytfa (15)—to include thermodynamic and metabolomics constraints (14) on top of enzyme-constrained models. Finally, we benchmark the assumptions and performance of the relaxation algorithms presented here and inspect the effects of the different layered constraints over the plain stoichiometric formulation, highlighting the difficulty and caveats of integrating different types of omics in the same model.

## MATERIALS AND METHODS

Two jupyter notebooks reproducing the results in this paper are provided as supplementary material: the S2 File corresponds to Fig. 1 to 3 and the S3 file corresponds to Fig. 4. The notebooks with their data ready to be reproduced are available at https://github.com/carrascomj/simult_supp_material.

### Flux balance analysis

FBA is a linear programming problem in the form of equation (5) (2).

$$\text{maximize } Z \tag{5}$$
$$\text{subject to } \boldsymbol{Sv} = 0 \tag{6}$$
$$lb_j \leq v_j \leq ub_j \tag{7}$$

where $\boldsymbol{Z}$ is the objective, $\boldsymbol{S}$ is the stoichiometric matrix. $\boldsymbol{v}$ is the vector of reaction fluxes, and $lb_j$ and $ub_j$ are the lower and upper bounds constraining the flux of the reaction variable $v_j$.

$\boldsymbol{Z}$ can be expressed as $\boldsymbol{c}^T\boldsymbol{v}$ where $\boldsymbol{c}$ is the coefficient vector that indicates the influence of each reaction in $\boldsymbol{v}$ over the objective. Typically, $\boldsymbol{c}$ is a standard basis

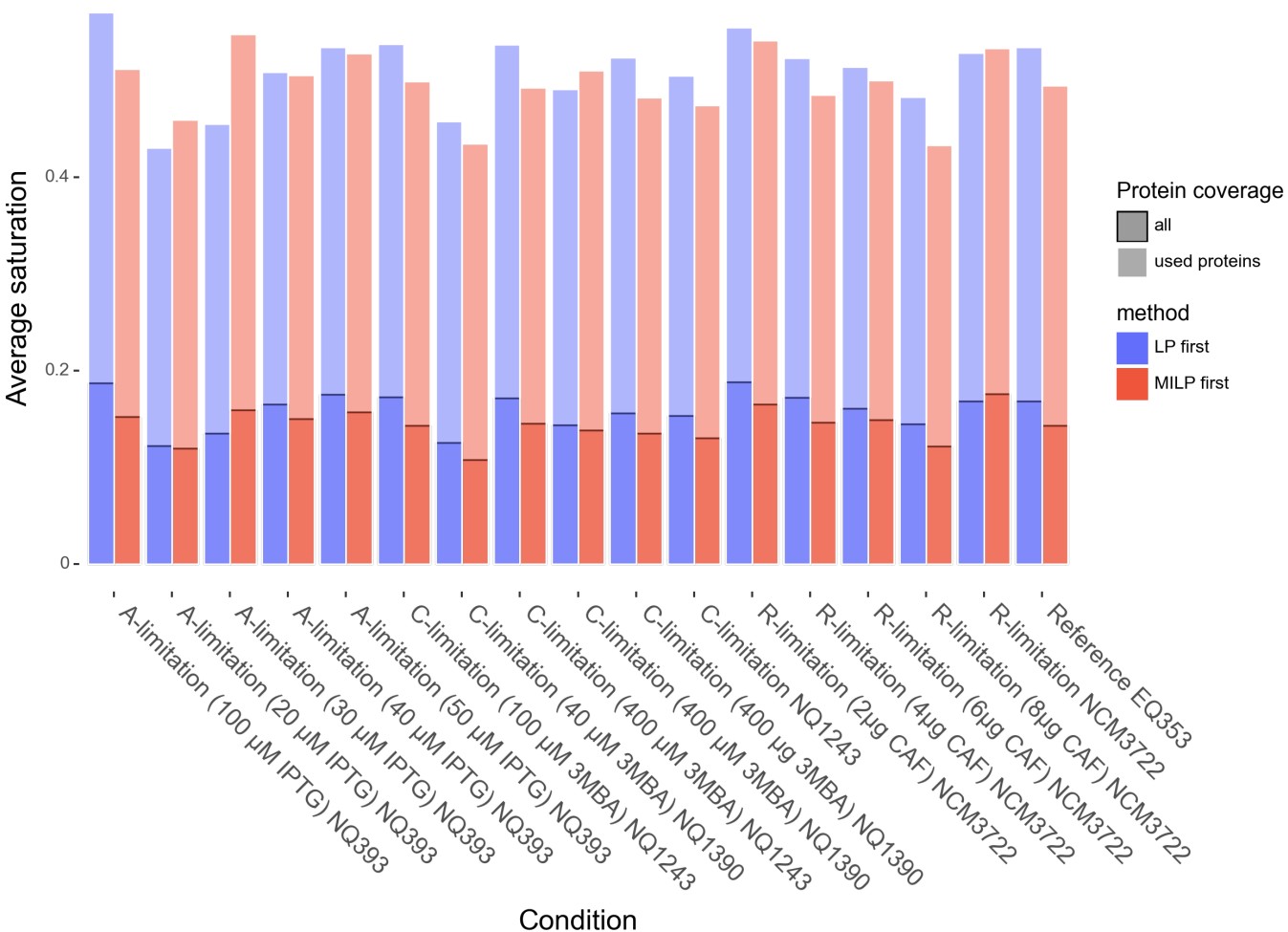

**FIG 1** Average enzyme saturation in terms of the protein usage (flux) of each protein divided by its concentration per condition. The opaque bars in front indicate the saturation accounting for all proteins while the longer transparent bars are the saturation only for active enzymes (primal value greater than $10^{-7}$).

vector referring to only one reaction, the biomass pseudoreaction or the ATP synthase 3. The biomass pseudoreaction acts as a sink in the metabolism, converting different byproducts of the metabolic network into cell mass (generally, grams of dry weight $g_{DW}$) yielding a growth rate ($1/h$), calibrated by experimental data.

Equation (6) constitutes the steady-state assumption, which forces a net balance of fluxes through the network. This produces an undetermined system of equations with infinite solutions (since there are more reactions than metabolites), whose solution space is further constrained by equation (7). The latter equation imposes the minimum and maximum flux value for each reaction in $v$. In big GEM models, the solution of fluxes that optimizes $\vec{z}$ is generally non-unique.

## Enzyme-constrained flux balance analysis

Enzyme-constrained flux balance analysis is another linear programming problem built on top of FBA. For a more detailed explanation, refer to references (12, 13). Briefly, the stoichiometric matrix is extended with protein exchanges (variables/columns) and protein species $P$ (constraints/rows).

In the enzyme-constrained formulation, a given enzyme $p$ participates in its respective reaction as a pseudometabolites with the stoichiometric coefficient $\frac{1}{k_{\text{cat},p}}$, where $k_{\text{cat},p}$ is the turnover number of protein $p$. The proteins are correspondingly

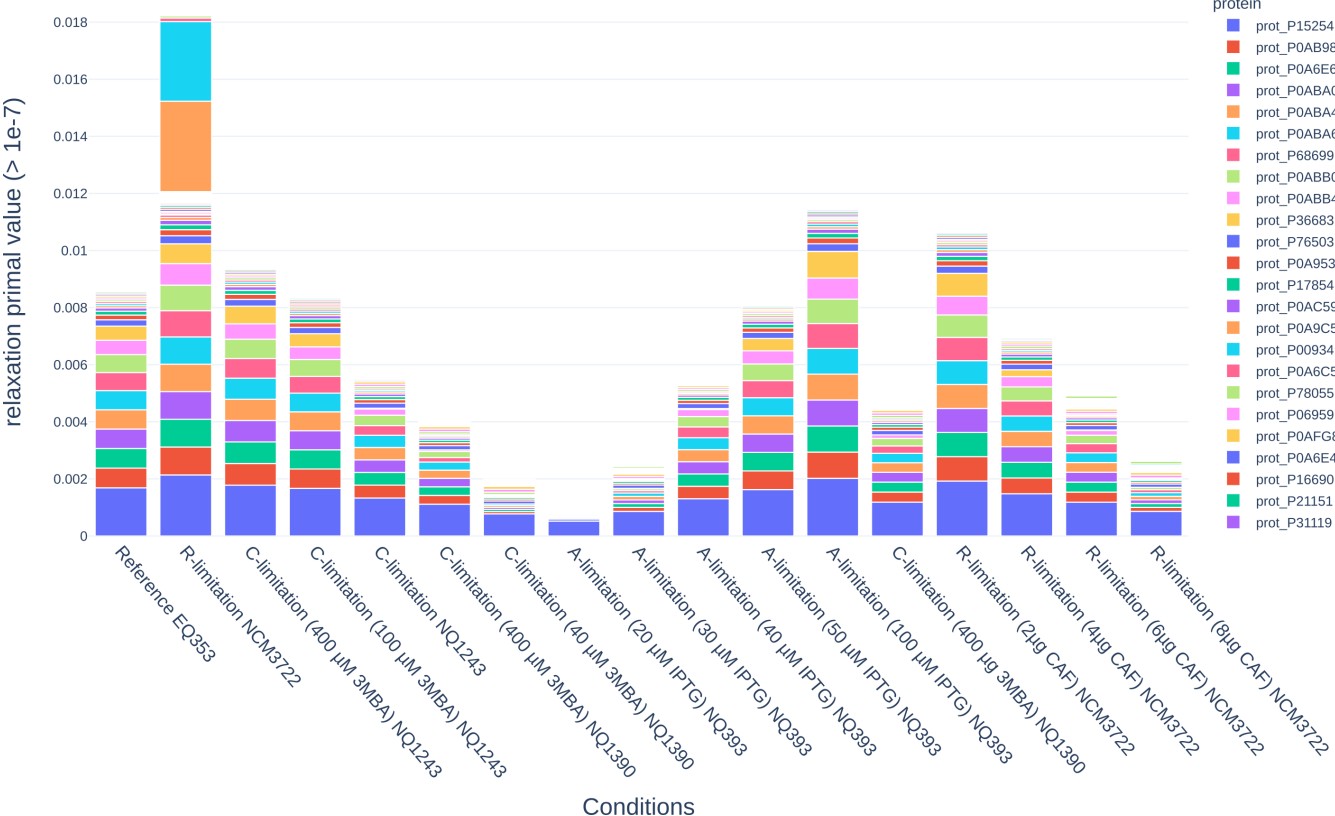

**FIG 2** Protein relaxation value (as primal values, in $\mathrm{mmol/g_{DW}/h}$) for the proteins of the irreducibly inconsistent set solved by the LP with one iteration per condition (LP first). Only relaxation values greater than $10^{-7}$ (in the numerical error threshold for the LP solvers) were reported.

supplied into the network by protein pseudoexchanges. It is in the upper bounds of these pseudoexchanges where the protein concentrations are expressed.

Furthermore, a global resource constraint can be applied to all or a subset of the proteins with unknown concentrations, commonly referred to as protein pool.

## Geckopy

A Python software package, geckopy, was developed building upon reference (12). The package was made open-source under the Apache2.0 license and it can be found at https://github.com/ginkgobioworks/geckopy, documented at https://geckopy.readthe-docs.io. The enzyme-constrained method stated above is on the core of geckopy, which derives from cobrapy (17) the API and provides an integration layer with pyTFA (15) to apply TFA on top of the enzyme-constrained models.

## Relaxation algorithms

A suite of relaxation algorithms was implemented to compute an irreducibly inconsistent set (IIS); that is, a minimal set of infeasible constraints. The IIS is resolved by the formulation of different LP or mixed-integer LP (MILP) problems that expand the original LP problem $S$ and may not be unique; therefore, a criterion is required to select one set. This corresponds to different objectives of the (MI)LP formulation. As noted in reference (19), these can be thought of as variations of minimization of metabolic adjustment (20). Three of these relaxation (MI)LPs were implemented:

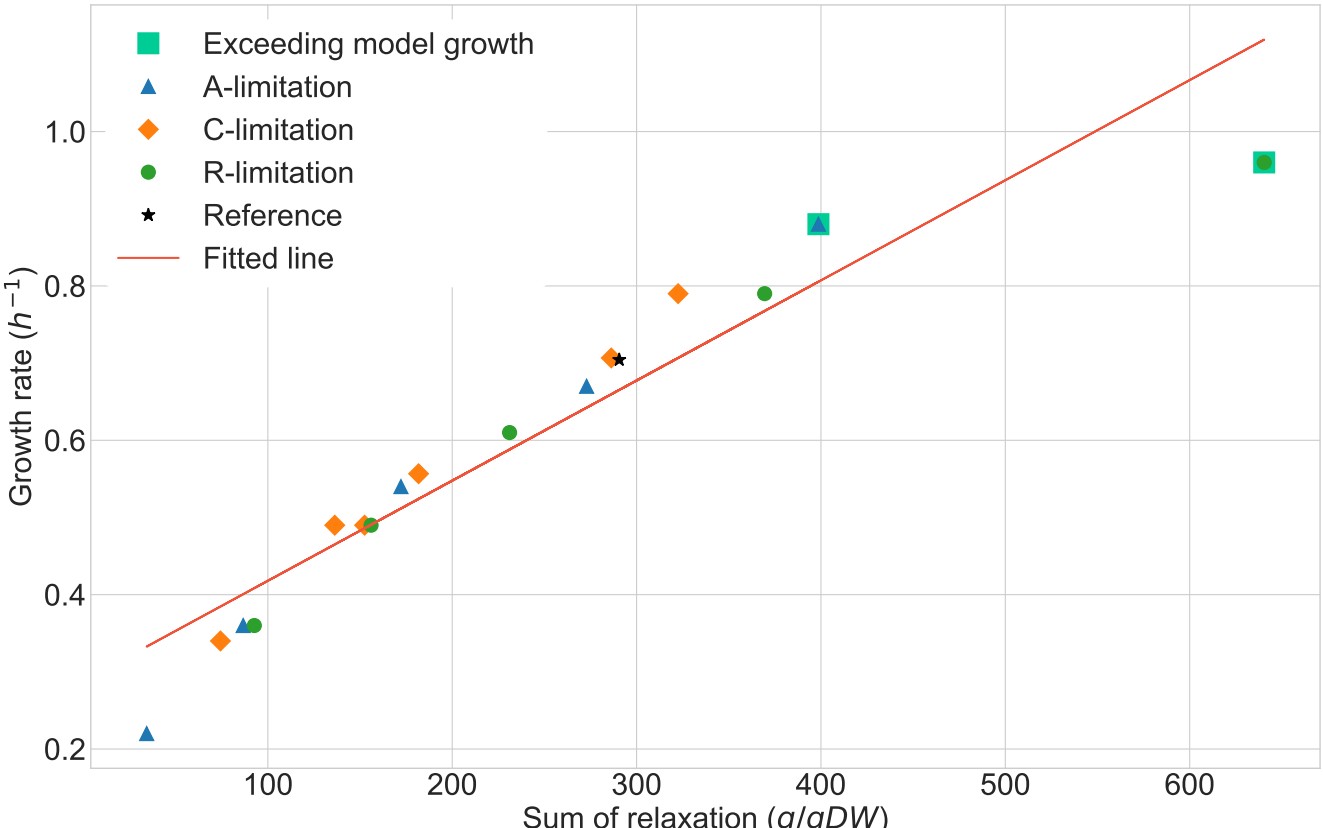

**FIG 3** Least squares fitting from the sum of the relaxation values (in g/gDW) for each proteomics condition with respect to the growth rate (h$^{-1}$). Pearson's correlation coefficient was 0.951. The squares represent the samples that exceeded the growth rate of—and that were capped to—the unconstrained growth of the model. The rightest condition, which deviates from the fitting, is the R-limited positive control.

- Elastic filtering algorithm as detailed in reference (21). Briefly, this is an LP problem that introduces elastic variables $e$ to a target set of constraints, one for each direction, to recover the feasibility of the solution. For this particular formulation, the target set of constraints are the enzyme pseudometabolites $\boldsymbol{P}$, in the upper direction, since the lower bound is always zero. Each constraint vector in $\boldsymbol{P}$ is denoted as $\boldsymbol{p}$. The criterion in this case is the sum minimizing the total value of the elastic variables $e$ as in equation (8).

$$
\begin{aligned}
\text{minimize} \quad & \sum_{e \,\in\, \text{elastic vars}} e \\
\text{subject to} \quad & \boldsymbol{Sv} = 0 \\
& -e + \sum_{v \,\in\, \boldsymbol{p}} v = 0, \quad \forall \boldsymbol{p} \in \boldsymbol{P} \\
& 0 \leq e \leq 1000 \\
& \text{lb}_j \leq v_j \leq \text{ub}_j
\end{aligned}
\qquad (8)
$$

- Equation (8) is run iteratively, removing the elastic variables that are found in the previous iteration as part of the optimized objective. This expands the IIS until no elastic variables can be added to obtain a feasible solution. Still, this IIS is neither deterministic nor unique and depends on the order in which the iterative sets are retrieved.

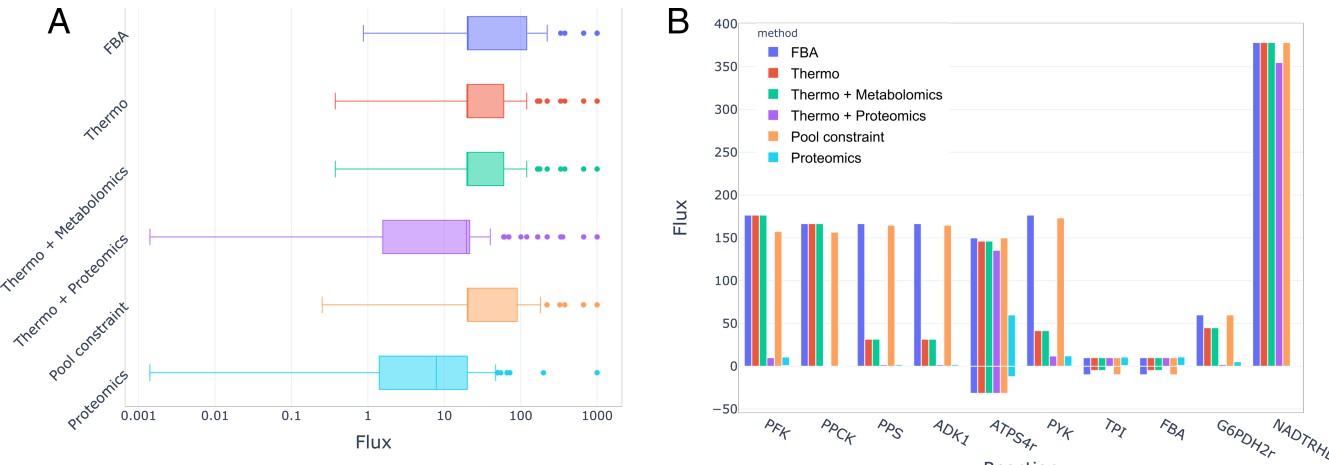

**FIG 4** (A) Distribution of non-zero intervals between maximums and minimums in flux variability analysis (FVA) of the *E. coli* core model, in logarithmic scale. Overall, proteomics constraints produce tighter distributions, with lower values of the interval. (B) FVA maxima and minima for some of the representative reactions in the model.

- Elastic filtering algorithm as with the previous case, changing the criterion to also optimize the original objective $Z$ of the problem as in equation (9).

$$
\begin{aligned}
\text{minimize} \quad & -Z + \sum_{e \in \text{elastic vars}} e \\
\text{subject to} \quad & Sv = 0 \\
& -e + \sum_{v \in p} v = 0, \quad \forall p \in P \\
& 0 \le e \le 1000 \\
& \text{lb}_j \le v_j \le \text{ub}_j
\end{aligned} \tag{9}
$$

- MILP problem minimizes the total count of variables in the IIS. This exploits that the relaxation is only in the upper bounds to avoid quadratic expressions. Thus, two variables are introduced for each enzyme constraint, an integer variable $i$ and a linear programming variable $l$ as in equation (10). Multiplying $i$ there is a large constant $K$ (1,000, the default upper bound for a COBRA reaction flux, sufficient enough to block the flux through a protein pseudoexchange reaction) which is compensated by $l$ when needed.

$$
\begin{aligned}
\text{minimize} \quad & \sum_{i \in \text{integer vars}} i \\
\text{subject to} \quad & Sv = 0 \\
& -i * K + l + \sum_{v \in p} v = 0, \quad \forall p \in P \\
& i \in \{0, 1\} \\
& 0 \le l \le K \\
& \text{lb}_j \le v_j \le \text{ub}_j
\end{aligned} \tag{10}
$$

In addition, a greedy brute-force algorithm was provided as it is implemented in Caffeine (22) or the GECKOmat suite (13). Briefly, this algorithm removes enzyme constraints iteratively guided by the shadow price—that is, the increment in the objective when one unit of the protein is increased—until the model reaches the growth specified by the user.

The different variants of formulations and objectives reflect different assumptions about the uncertainty of the experimental methods in place. Hence, if it is suspected that the uncertainty is uniformly distributed over all measurements, one of the elastic

filtering methods should be chosen over the MILP problem. Correspondingly, if the *a priori* knowledge points to a reduced subset of the enzymes with high uncertainty, the MILP might be a better fit. In addition, the original objective can be included in the relaxation problem. This may be useful when the model is constrained to a known experimental growth rate but the relaxation fails to find a solution with that constraint.

## Proteomics relaxation benchmark

To test the relaxation in absolute proteomics, the relaxation algorithms implemented in geckopy were benchmarked against a data set of absolute proteomics across different conditions in *E. coli*, retrieved from reference (23). The input of the absolute proteomics data sets, $\phi$, is defined in equation (11)

$$\phi_i = \frac{\mu_i I_i}{\sum_{k=1}^{\text{proteins}} \mu_k * I_k} \tag{11}$$

which is the protein mass fraction of each protein $i$, given the intensity $I_i$ of a quantitative proteomic approach (e.g., xTop, TopPep1/3, and iBAQ) and molecular mass $\mu_i$.

Each $\phi_i$ was converted to the units used by geckopy $\frac{\text{mmol}}{\text{g}_{DW}}$, by equations (12) and (13).

$$\frac{\text{mmol}}{\mu m^3} = \phi_i \frac{13.5 \times 10^{-8} \mu g}{\mu m^3} \frac{1g}{10^6 \mu g} \frac{1000 \cdot \text{mmol}_i}{M_{w,ig}} \tag{12}$$

$$\frac{\text{mmol}}{\text{g}_{DW}}_i = \frac{\text{mmol}}{\mu m^3}_i \frac{X \mu m^3}{OD \cdot mL} \frac{OD \cdot mL}{320 \mu g \cdot protein} \frac{10^6 \mu g \cdot protein}{g \cdot protein} \frac{0.448g \cdot protein}{\text{g}_{DW}} \tag{13}$$

$$X = -0.3 \times 10^9 GR + 2.83 \times 10^9 \tag{14}$$

Here, as derived empirically in the supplementary material of reference (24), $13.5 \times 10^{-8}$ is a constant, X is a linear fit of the cellular volume per OD in terms of the growth rate (GR, 14) as described ; 320 $\mu$g is the average number of proteins per ODmL for *E. coli* (25); and g/g$_{DW}$ was set to 0.448 for all conditions. The latter is an empirical constant although the actual value (or a growth-rat- based fit like 14) for the particular experimental conditions would be a more accurate approach. The only carbon source uptake, glucose, was set to 10 mmol/gDW · h, for the lack of an experimental value for this parameter. The growth rate was constrained to the upper limit of the confidence interval at 95 reported in the experimental conditions or the optimum growth rate for the FBA (non-enzyme-constrained) problem.

These constraints were enforced across all 17 conditions, using the upper limit of the confidence interval at 95 for concentrations and the lower limit for growth conditions where biological replicates were available. An IIS was solved for all of them individually with the MILP approach and the LP approach, both with and without expanding the IIS. The chosen objective was to minimize the sum of relaxation variables (count in the case of LP) without the growth rate since the model is constrained to this value. The GEM model of choice was the enzyme-constrained version of iML1515 (26) and eciML1515 (13).

Conditions consist of different strains constructed to modulate three substrate limitations: ammonia limitation (A), carbon limitation (C), and ribosome limitation (R). As explained in Table 1, just one modification was done to the model, in the case of ammonia limitation, where the GLUDy reaction was knocked out. The rest of the modifications were assumed to be accounted for by proteomics data since they refer to the modulation of the expression of either a protein or the whole proteome (R-limitation). The implementation is found in the S2 File, *proteomics_data_relaxations.ipynb*.

**TABLE 1** Modeling details of the different strains and conditions for the proteomics data sets of reference (23)[a,b]

| Strain ID | Genes | Description | Modeling |
|---|---|---|---|
| NCM3722 | K12 WT | K12 WT | eciML1515 |
| EQ353 | NCM3722, KD pyrE | - | KD pyrE |
|  |  |  | Accounted in proteomics |
| NQ1243,NQ1390 | NCM3722, Pu-PtsG | C-limitation, Glc-uptake regulated by 3MBA | PtsG accounted in proteomics |
| NQ393 | NCM3722, ΔgdhA, $P_{Llac}$-gltBD | A-limitation, GOGAT controlled by IPTG | KO GLUDyNo1,KO GLUDy_REVNo1 |
| - | - | R-limitation, chloramphenicol inhibits translation | Accounted in proteomics |

[a]KD stands for knock-down. Pu is a *Pseudomonas putida* promoter inserted in Pstd to 230 control it *via* 3MPA.
[b]The (MI)LP problems were solved using CPLEX (27).

## Flux variability analysis comparison

The layered constraints—stoichiometric, thermodynamics, metabolomics, proteomics, and pool assumption—were compared using flux variability analysis (FVA).

FVA is a COBRA algorithm that first runs an FBA problem and fixes the previous objective to its objective value. Then, for each reaction $j$, an FBA problem is formulated where the objective $Z$ corresponds to the flux through the $j$; more formally, $Z = e_j^T \boldsymbol{v}$. This problem is optimized for both the minimization and maximization directions. The output is the maximum and minimum possible flux of each reaction in the model. Thus, the distribution of maximum and minimum fluxes of the solution space for a particular objective can be inspected using this method.

To ease the interpretation of this analysis, a reduced GEM model was used. Enzyme variables were added to the *E. coli* core model from BiGG (28), with 95 reactions, 72 metabolites, and 55 proteins, to form a reduced enzyme-constrained model. This model is available in the S1 File. The model was built by taking the subset of reactions in the eciML1515 model and transferring its enzyme variables and turnover numbers to the reduced *E. coli* core model.

For the sake of a fair comparison, the glucose exchange was fixed to its minimum possible value inside the optimum biomass solution before running the FVA (13). Although an FVA implementation that blocks the opposite splitted reaction (common in EC models) to remove spurious variability was implemented as part of geckopy, this model does not have splitted reactions. Moreover, the model does not have isozymes, which further simplifies the analysis. The growth was enforced to be equal to the thermodynamic solution, which did not require relaxation to satisfy a feasible solution.

The relaxation of experimental data used was always the IIS MILP (equation (10)) + the greedy filter over the IIS, which minimizes the number of constraints removed from the problem.

## Average enzyme saturation

The average enzyme saturation was calculated as implemented in geckopy, following equation (15)

$$E_{\text{sat}} = \frac{1}{n} \sum_{v \in \boldsymbol{v_p}} \frac{v}{\text{ub}_v} \tag{15}$$

where $\boldsymbol{v_p}$ is the vector of protein exchange variables whose upper bound is non-zero, $ub_v$ is the upper bound of the given variable; and $n$ is the length of $\boldsymbol{v_p}$.

## RESULTS

### Geckopy

#### Design principles

Three design principles were chosen for the development of geckopy:

1. SBML compliance, to maintain the community standards.

2. API separation of the proteins from metabolites and reactions.

3. Ease of experimental data loading.

#### SBML specification

As in previous iterations of enzyme-constrained models (12, 13), proteins are *SBML Species* (29). This change moves away from naming conventions to identify enzymes with a structured system that more appropriately complies with the FAIR principles (30). Here, we proposed to extend this definition to enzymes as *Species*:

- in a defined SBML group (4) named "Protein";

- with the concentration specified as the optional Species attribute *initialAmount*; and

- whose $k_{cat}$ value is specified in the stoichiometry coefficient of the reactions, as usual.

The reasoning behind this specification is to ensure that the document is self-contained and does not depend on naming conventions during the parsing. Thus, modelers can save in the document proteomics information if needed and the SBML standard is preserved.

Listing 1: SBML document with proteins.

```
<listOfSpecies >
   <species id="P00363" compartment="c" initialAmount="0.12"/>
   <species id="P08921" compartment="c" initialAmount="1.4"/>
</listOfSpecies>
<listOfGroups
xmlns="http://www.sbml.org/sbml/level3/version1/groups/version1">
   <group kind="classification" id="Protein">
      <listOfMembers id="all_proteins">
         <member idRef="P00363"/>
         <member idRef="P08921"/>
      </listOfMembers>
   </group>
</listOfGroups>
```

Geckopy implements the GECKO algorithm (12). Enzyme-constrained models expand the stoichiometry matrix of the classic flux balance analysis formulation (2)—designated as $S$—with enzyme constraints—designated as $P$. The proteins are supplied by variables with an upper bound that reflects the experimental measure when known, that is, pseudoexchanges that supply the protein to be consumed in the reactions; and appear as constraints in the reaction variables where they participate, that is, pseudometabolites with a stoichiometric coefficient as the inverse $k_{cat}$. Geckopy implements this, built on top of the established COBRA software cobrapy (17). In addition, geckopy includes a suite of flux analysis algorithms for enzyme-constrained models analogous to the ones in cobrapy, such as flux variability analysis, parsimonious flux protein analysis, etc.

Another difference with the GECKO toolbox in MATLAB (13) is that geckopy implements the reversible reactions internally, removing the need to split all of the reversible

reactions in the SBML. Nonetheless, geckopy can read *legacy* enzyme-constrained models that use splitted reactions and/or proteins specified with the naming convention *prot_UNIPROTID*.

## Benchmark of relaxation algorithms with proteomics data sets

The relaxation of experimental constraints is usually a requirement to yield a feasible enzyme-constrained model. To test the assumptions and show the utility of the relaxation algorithms suite, they were benchmarked against a data set of *E. coli* growing in 17 conditions (carbon, ammonia, and ribosome limitation) from reference (23).

For each condition, the eciML1515 model (4,824 reactions, 2,333 metabolites, and 1,259 enzymes) was constrained with the reported protein measurements and the experimental growth rate (see Materials and Methods). Of the 1,259 enzymes in the model, 1,243 were found in the experimental data consistently across conditions. As detailed in Table 2, across samples, the MILP required a relaxation of more proteins than the LP counterparts. The worst performer by an order of magnitude was the greedy algorithm.

By contrast, the average enzyme saturation in Fig. 1 for the LP first and the MILP first methods was similar across samples (below 20% for all enzymes and around 50% if only active enzymes are accounted for), but with an overall decrease (less usage) for the MILP case.

For the LP cases, inspecting the relaxation values of the proteins—that is, how much the experimental measurement must be violated—is interesting to assess possible errors in the model or the experimental data. This informs not only about a protein being relaxed but about how much it was relaxed. A closer look at the relaxation values in the LP first case in Fig. 2 revealed that the number of proteins in the IIS drops down to 74.05 when taking into account only primal values greater than $10^{-7}$ (which is on the threshold for numerical errors). Moreover, 10 proteins hold around 77.12 % and 90.97 % of the total relaxation of the problem across the samples. These proteins were the phosphoribosylformylglycinamidine synthase, the aconitase, subunits of the ATP synthase, and the 3-oxoacyl-[acyl-carrier-protein] synthase 1. One deviation from this trend was the R-limitation positive control, which additionally showed two proteins with a large relaxation value: a phosphoserine phosphatase and a riboflavin reductase.

As expected, it should be noted in Fig. 2 that the growth is highly correlated—Pearson's correlation coefficient of 0.951 (Fig. 3)—with the sum of the relaxation values. This indicates that the enzyme constraints are less important when moved away from optimal growth conditions, whereas uptake rates dominate the solution in these cases. Once in optimal conditions, the higher relaxation values may be highlighting an overall underestimation of $k_{cat}$ values that were not observed in lower growth rates, when operating under non-saturation rates.

## Layered comparison of constraints

Geckopy implements an integration API with pytfa (15) to add thermodynamic and metabolomics constraints on top of the enzyme-constrained models. Thus, it was relevant to check whether the solution space becomes smaller—and thus less uncertain—when applying combinations of these different constraints.

Using a reduced model of *E. coli* comprising mainly the glycolysis, pentose phosphate pathway, and the tricarboxylic acid cycle, the comparison of FVA runs shows the effect

**TABLE 2**  Number of enzymes relaxed per condition

| Method | Mean | Standard deviation |
|---|---|---|
| Greedy | 791.529 | 40.928 |
| LP | 74.000 | 11.853 |
| LP first | 75.823 | 12.826 |
| MILP | 95.000 | 11.296 |
| MILP | 95.048 | 11.305 |

of the different constraints layered in the reduction of the solution space in Fig. 4. The relaxation for the "Thermodynamics + Proteomics" scenario removed 22 proteins, and the "Proteomics" removed 18. It can be observed in Fig. 4A that the flux interval distribution for the proteomics constraint is less dispersed, followed by the thermo and proteomics constraints, the thermodynamic constraints, the pool constraint, and, finally, the flux balance analysis. This is reflected in Fig. 4B, where it can be seen how the different constraints can turn a reversible reaction into irreversible (TPI, proteomics), reduce the flux allowed, fix it to a reduced range (PPS, proteomics), or block it completely (PPCK).

## DISCUSSION

### The relaxation of the enzyme-constrained model reveals inconsistencies between the model and the experimental measurements

The exploration of the IIS of several proteomics data sets revealed 10 proteins that are significantly and consistently clashing between the model and the data. Of these proteins, six are subunits of the ATPase, which is not only a transmembrane protein—which can pose quantification challenges to proteomics methods—but also a protein complex with complex kinetics where the enzyme-constrained assumption may not hold as well as with the usual metabolic enzymes, since a $k_{cat}$ cannot summarize the mechanochemical operation of the ATPase (31). Among the other four, we found $k_{cat}$ values in the first percentile of the $k_{cat}$ values in the model, indicating possible underestimations of the actual *in vivo* $k_{cat}$ values by *in vitro* methods.

This information, consistent across samples and experimental methods, could be used to flag regions where the protein coverage of the model can be improved, kinetic parameters updated, or where experimental methods may lead to measurement errors. There were two outliers in the R-limitation (translation inhibition) positive control condition: the phosphoserine phosphatase and one isozyme of the FMN reductase. These two enzymes also have comparably low $k_{cat}$ values and are responsible for the total relaxation being higher than expected for the growth rate of the condition (rightmost point in Fig. 3). It is important to note that this condition exhibited the highest growth rate, exceeding the growth of the model, so it is possible that the experimental glucose uptake rate also exceeded the modeled value of $10 \text{ mmol/gDW/h}$, impacting the relaxation.

Overall, the number of relaxed enzyme constraints in the MILP problems was greater than the LP counterparts overall, producing less constrained problems as represented by the decrease in average enzyme saturation. Therefore, it may be preferable to choose the LP elastic relaxation instead, which only adjusts the experimental measurements, instead of the MILP, which removes them. Moreover, the LP formulation has the advantage of providing relaxation values for each relaxed protein, providing a measure of the extent of the violation of the experimental data.

### Layered comparisons of constraints reveal inconsistencies when combining experimental constraints of different nature

As it was reported in Fig. 4, enzyme constraints result in a more constrained space solution than in the thermodynamic constraints. This was expected, as 47 reactions (49.47%) in the core model are already unidirectional, which is the main source of constraints that TFA is expected to enforce (14).

It was less expected that the proteomics constraints alone would produce a smaller solution space than the enzyme constraints + thermodynamics. However, this is clearly explained by looking at the number of proteins removed from the problem in the two cases. This indicates that absolute proteomics constraints introduce inconsistencies between the experimental measurements and the model which may arise from (i) inaccurate experimental measurements, (ii) lack of coverage of isozymes and promiscuity in the model, and (iii) lack of coverage of reactions in the model. In this case, the model

presented is a reduced reconstruction, which is likely the greater source of inconsistencies.

It is worth noting that this behavior of an increased solution space for combinations of thermodynamics and enzyme constraints had been previously reported (19). Furthermore, better management of uncertainty and the use of more accurate methods to estimate free energies, like component contribution, could provide more informative thermodynamics constraints that enhance the predictions, as in reference (32).

## Conclusion

In this work, we have presented the Geckopy 3.0a package that, beyond the enzyme-constrained formulation, provides three innovations:

1. A way to include proteins as first-class citizens in the SBML document, complying with the community standards, with its corresponding parser implementation.

2. Inspection and biological discussion of a suite of relaxation algorithms, inspired by reference (19), which may be used to reveal inconsistencies in the model or the proteomics data if used systematically.

3. An integration with pytfa (15) to include thermodynamic and metabolomics constraints on top of the enzyme constraints. The inspection of this integration revealed that combining both proteomics and metabolomics may not provide a more constraint space than one of them alone, in accordance with previous works in the field (19).

In future work, the inclusion of multiTFA (32) instead of pytfa and more sophisticated approaches to combine both enzyme-constrained models and TFA might be explored. Overall, we would like to emphasize the need in the field to integrate multi-omics methods consistently.

## ACKNOWLEDGMENTS

We would like to acknowledge Benjamín Sánchez and Iván Domenzain for their valuable perspectives and discussions on enzyme-constrained models.

We would also like to thank the Ginkgo Bioworks team who kindly spent time hearing about geckopy during its development and offering feedback. J.C.M. would also like to thank Shannara Taylor Parkins and Teddy Groves for their advice during the review.

N.S. acknowledges funding from the Novo Nordisk Foundation under the Fermentation-Based Biomanufacturing program (grant no. NNF17SA0031362). J.C.M. acknowledges funding from the Novo Nordisk Foundation (grant no. NNF20CC0035580).

J.C.M. declares no conflicts of interest. C.L. and N.S. are employees of Ginkgo Bioworks.

## AUTHOR AFFILIATIONS

[1]Department of Biotechnology and Biomedicine, Technical University of Denmark, Kongens Lyngby, Denmark
[2]Novo Nordisk Foundation for Biosustainability, Technical University of Denmark, Kongens Lyngby, Denmark
[3]Ginkgo Bioworks, Boston, Massachusetts, USA

## PRESENT ADDRESS

Nikolaus Sonnenschein, Ginkgo Bioworks, Boston, Massachusetts, USA

## AUTHOR ORCIDs

Jorge Carrasco Muriel ⓘ http://orcid.org/0000-0001-7365-0299

## FUNDING

| Funder | Grant(s) | Author(s) |
|---|---|---|
| Novo Nordisk Fonden (NNF) | NNF17SA0031362 | Nikolaus Sonnenschein |

## ADDITIONAL FILES

The following material is available online.

### Supplemental Material

**S1 file (Spectrum01705-23-s0001.xml).** E. coli reduced genome-scale network reconstruction.
**S2 file (Spectrum01705-23-s0002.html).** Jupyter notebook to reproduce figures 1, 2, 3.
**S3 file (Spectrum01705-23-s0003.html).** Jupyter notebook to reproduce figure 4.

### Open Peer Review

**PEER REVIEW HISTORY (review-history.pdf).** An accounting of the reviewer comments and feedback.

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
