## [Reviewer comments · Microbiology Spectrum]

Microbiology Spectrum

Simultaneous application of enzyme and thermodynamic constraints to metabolic models using an updated Python implementation of GECKO

Jorge Carrasco Muriel, Christopher Long, and Nikolaus Sonnenschein

Corresponding Author(s): Jorge Carrasco Muriel, Danmarks Tekniske Universitet The Novo Nordisk Foundation Center for Biosustainability

Review Timeline:

Submission Date:	April 26, 2023
Editorial Decision:	June 2, 2023
Revision Received:	August 15, 2023
Editorial Decision:	September 5, 2023
Revision Received:	September 11, 2023
Accepted:	September 11, 2023

Editor: Angela Re

Reviewer(s): Disclosure of reviewer identity is with reference to reviewer comments included in decision letter(s). The following individuals involved in review of your submission have agreed to reveal their identity: Ivan Domenzain (Reviewer #1)

Transaction Report:

DOI: <https://doi.org/10.1128/spectrum.01705-23>

June 2, 2023

Dr. Jorge Carrasco Muriel
Danmarks Tekniske Universitet The Novo Nordisk Foundation Center for Biosustainability
Copenhagen
Denmark

Re: Spectrum01705-23 (Simultaneous application of enzyme and thermodynamic constraints to metabolic models using an updated Python implementation of GECKO)

Dear Dr. Jorge Carrasco Muriel:

Link Not Available

Sincerely,

Angela Re

Journals Department
Reviewer comments:

Reviewer #1 (Comments for the Author):

Major concerns:

1.- The geckopy implementation presented here is a software pipeline for integration of omics and thermodynamic constraints into an enzyme-constrained model of metabolism. This is well explained by the title. However, the abstract and introduction sections do not specify that the enzymatic constraints, or catalytic constants are not treated here and are taken as inputs in the geckopy pipeline. The current text may sound self-explanatory to the authors or very specialized researchers, but the readers of Microbiology Spectrum include a broader audience.

2.- The tool is presented as a version 3.0 of the originally proposed geckopy by Sánchez, et al. 2017. At the same time it is highlighted that this is a reconstruction from scratch, which implies independent development, which I do agree that is presented here, but breaks strict software versioning practices. A problem, intrinsic to GECKO available at:

<https://github.com/SysBioChalmers/GECKO>, is that development of the python module stopped practically at its first version, whilst MATLAB versions presented methodological changes in data integration (GECKO 2.0), and even at the model format level (GECKO 3.0). Therefore, it is strictly recommended to be consistent with the title of this manuscript and refer to the new tool as an independent implementation of data constraints in GECKO throughout the rest of the text and associated materials.

3.- Across the whole manuscript there are several terms that differ from the ones that have been used in the studies of this modeling field. In particular, the term enzyme constraint model is repeatedly used in this manuscript. The habit has been to name these models as enzyme, enzymatically or protein models. Consistency with the terminology of the subfield is recommended unless a re-discussion of the term is presented.

4.- The manuscript and supplementary materials do not provide enough detail the parameterization procedure of the reduced metabolic model (i.e. selection of k_{cat} values). It has been reported by several studies that k_{cat} distributions play a major role in flux distributions, therefore, this factor would be expected to majorly impact the conclusions extracted from the results of the flux variability analysis results. It is necessary to provide such parameterization details and complement discussion with more details regarding the selection of parameters. Particularly those discussed at the reaction level. A property of enzyme constrained models is that systems-level distribution of kinetic parameters translates to changes at the reaction level, and even though this is a reduced model, the number of reactions and metabolites suggest that it represents the conjunction of multiple pathways. It is also recommended to mention the sectors of metabolism that this small models accounts for, in order to highlight the relevance of conclusions to a wider public, such as non-modeler microbiologists.

5.- The average enzyme saturation across enzymes is analyzed at a genome-scale by using different relaxation algorithms. This analysis returned an overall low saturation value as all individually constrained enzymes were taken into account. Looking separately at the number of "used" enzymes (the flux carrying ones) may offer an additional factor for comparison, and also calculating a separated saturation factor for this group of enzymes may help the discussion of the commonly adopted average saturation factor of 0.5 in models constrained at the protein pool level.

6.- It is not clear in the manuscript and methods how the carbon source uptake rate is used as a constraint in addition to those at the proteomic level. This information is of particular importance for constraining a metabolic network, as it is indicative of the metabolic state, mode and even stress levels for a given growth phenotype.

7.- Comparison of the proposed LP and MILP algorithms for selection of IIS to the brut-force algorithms available in GECKOmat and caffeine is lacking and is strongly recommended, for the sake of providing quantitative evaluation of approaches to the interested community. When referring to the LP approach in the benchmarking section it is not clear which of the two methods presented for selection of IIS was used (optimization of elastic variables or elastic variables + objective). As growth rate is used as a constraint, this may suggest the former, but not explicitly said and also not clear for researchers starting to explore the subfield.

8.- What does a "a large constant K" (line 169) means in the context of the MILP problem. How large is it? what was the rationale behind the chosen value? what is the impact of this parameter in the performance of the MILP-based approach?

9.- Table 1 is not informative enough of the modifications, condition parameters and assumptions used to model the conditions of origin for the different proteomic datasets.

10.- Provide statistical tests for the comparison of flux variability ranges across different layers of constraints and add significance arguments to the discussion of these results

Minor comments:

11.- Line 24: replace "Genome-scale model Enzyme constraints" by "Genome-scale model with Enzyme constraints" as originally named by Sánchez, et al., 2017.

12.- Line 65: "Now by enforcing mass conservation over the network, $S_v=0$ ". The explanation is not as straightforward as mentioned here and has been extensively described in other specialized reviews. Better cite those and rephrase.

11.- Line 76: "an update of an existing open-source enzyme-constraint software: geckopy 3.0" please follow the recommendation in the major concerns and be more descriptive here, the presented geckopy is a software for incorporation of omics constraints into models with enzymatic parameters.

12.- Line 77: "a SBML" acronyms starting with "S" are usually referred to with "an".

13.- Line 78: recommended to call proteins as "pseudometabolites" in order to avoid confusion.

- 14.- Line 112: "v is the vector of reaction variables". More than 20 years of FBA related papers have named this as a vector of reaction fluxes, recommended to indicate that the prediction outcome is the distribution of fluxes as a separate idea.
- 15.- Line 116: Provide references to the use of the biomass pseudoreaction or the ATP synthase as objective functions.
- 16.- Line 154: "a LP problem", acronyms starting with L are usually referred to with "an".
- 17.- Line 159: "the total flux", not clear if the authors refer to the metabolic flux that those enzymes carry or if they mean total use of elastic variables instead.
- 18.- There is a gap in between lines 163 and 164 that does not clarify what the listed points are specifically referring to.
- 19.- Line 225: not clear with what is meant with "when optimizing biomass for all reactions" in this context.
- 20.- A one line explanation describing the thermodynamic solution will help to clarify this section.
- 21.- Line 298: The term the number of conditions is not clear when reading this and also figure 1 with its corresponding caption. What does it mean? Do the authors refer to the identifiers of the conditions/samples (e.g. sample 1, sample 2, ..., sample 17)? Clarifying this will majorly help this section.
- 22.- The average number of relaxed proteins across algorithms reported in lines 299- 302 does not seem to correspond with the counts values shown in figure 1 (y-axis). Or do they refer to different variables?
- 23.- Line 317: Here fig. 4 is referred to before introducing figure 3, therefore the order of these figures should be interchanged so that it reflects the sequence in the text.
- 24.- Line 322: typo in "one deviation form this trend"
- 25.- Current figure 4 shows growth rate sum of relaxation values, but units are not reported for any of the axis. According to equation 8, I assume that the sum of relaxation values is presented in mmol/gDw (such as enzyme usages). It is recommended to convert the sum of relaxation values to mass units, by multiplying the contribution of every flexibilized enzyme by its molecular weight. In this way it is possible to assess what is the proportion of flexibilized data in comparison to the total protein content of the cell (global constraint) in terms of a conserved quantity.
- 26.- Additionally, I recommend to add a color code or different markers to the data points in figure 4, indicating the modeled conditions, as the association between growth rates and experimental conditions is not provided anywhere else in the text or associated materials. This will help top clarify what do authors mean with expression such as "protein constraints are less important when moved away from optimal conditions" (line 327).
- 27.- Line 334: the sentence "... samples that exceeded the growth rate of and what were capped ..." is hard to understand, probably missing words between "for" and "and".

Reviewer #2 (Comments for the Author):

See comments in attached pdf

Staff Comments:

Preparing Revision Guidelines

Please return the manuscript within 60 days; if you cannot complete the modification within this time period, please contact me. If you do not wish to modify the manuscript and prefer to submit it to another journal, please notify me of your decision immediately so that the manuscript may be formally withdrawn from consideration by Microbiology Spectrum.

The manuscript " Simultaneous application of enzyme and thermodynamic constraints to metabolic models using an updated Python implementation of GECKO" provides a python-based software implementation for facilitated integration of proteomics and thermodynamic constraints into metabolic models. A genome scale enzyme-constrained model of the metabolism of Escherichia coli is used to test the different data flexibilization/reconciliation algorithms available in the software. Additionally, a reduced model of E. coli's metabolism is used to assess the reduction in the solution space (allowable predictions) by the introduction of different combination of constraints (thermodynamic and proteomic, with and without addition of omics measurements at the molecular level).

The presented study offers substantial methodological advancements to the field of metabolic modeling of microorganisms, in particular to the subfield of enzyme constraints, which is undergoing recent significant developments. Namely, the authors improve the functioning and systematized the incorporation of proteomics constraints into metabolic models by totally free open source software; provide further compatibility of this kind of models and omics data with the SBML standard and the widely used simulation tool COBRApy; demonstrate how the study of proteomics data across diverse conditions, together with other layers of constraints and contextualized metabolic knowledge, contribute to pinpoint model and/or data inconsistencies at the reaction and protein levels.

Overall, this is a high-quality research product that offers tools and insights of relevance for publication in Microbiology Spectrum, as it touches upon two of the points listed in its scope (Findings that are of primary interest to smaller sub-fields within microbiology, and re-analyses of large datasets that provide additional insights). I recommend accepting this manuscript for publication after the following major and minor points of concern have been clarified and considered.

Major concerns:

1.- The geckopy implementation presented here is a software pipeline for integration of omics and thermodynamic constraints into an enzyme-constrained model of metabolism. This is well explained by the title. However, the abstract and introduction sections do not specify that the enzymatic constraints, or catalytic constants are not treated here and are taken as inputs in the geckopy pipeline. The current text may sound self-explanatory to the authors or very specialized researchers, but the readers of Microbiology Spectrum include a broader audience.

2.- The tool is presented as a version 3.0 of the originally proposed geckopy by Sánchez, et al. 2017. At the same time it is highlighted that this is a reconstruction from scratch, which implies independent development, which I do agree that is presented here, but breaks strict software versioning practices. A problem, intrinsic to GECKO available at: <https://github.com/SysBioChalmers/GECKO>, is that development of the python module stopped practically at its first version, whilst MATLAB versions presented methodological changes in data integration (GECKO 2.0), and even at the model format level (GECKO 3.0). Therefore, it is strictly

recommended to be consistent with the title of this manuscript and refer to the new tool as an independent implementation of data constraints in GECKO throughout the rest of the text and associated materials.

3.- Across the whole manuscript there are several terms that differ from the ones that have been used in the studies of this modeling field. In particular, the term enzyme constraint model is repeatedly used in this manuscript. The habit has been to name these models as enzyme, enzymatically or protein models. Consistency with the terminology of the subfield is recommended unless a re-discussion of the term is presented.

4.- The manuscript and supplementary materials do not provide enough detail the parameterization procedure of the reduced metabolic model (i.e. selection of k_{cat} values). It has been reported by several studies that k_{cat} distributions play a major role in flux distributions, therefore, this factor would be expected to majorly impact the conclusions extracted from the results of the flux variability analysis results. It is necessary to provide such parameterization details and complement discussion with more details regarding the selection of parameters. Particularly those discussed at the reaction level. A property of enzyme constrained models is that systems-level distribution of kinetic parameters translates to changes at the reaction level, and even though this is a reduced model, the number of reactions and metabolites suggest that it represents the conjunction of multiple pathways. It is also recommended to mention the sectors of metabolism that this small models accounts for, in order to highlight the relevance of conclusions to a wider public, such as non-modeler microbiologists.

5.- The average enzyme saturation across enzymes is analyzed at a genome-scale by using different relaxation algorithms. This analysis returned an overall low saturation value as all individually constrained enzymes were taken into account. Looking separately at the number of “used” enzymes (the flux carrying ones) may offer an additional factor for comparison, and also calculating a separated saturation factor for this group of enzymes may help the discussion of the commonly adopted average saturation factor of 0.5 in models constrained at the protein pool level.

6.- It is not clear in the manuscript and methods how the carbon source uptake rate is used as a constraint in addition to those at the proteomic level. This information is of particular importance for constraining a metabolic network, as it is indicative of the metabolic state, mode and even stress levels for a given growth phenotype.

7.- Comparison of the proposed LP and MILP algorithms for selection of IIS to the brut-force algorithms available in GECKOmat and caffeine is lacking and is strongly recommended, for the sake of providing quantitative evaluation of approaches to the interested community. When referring to the LP approach in the benchmarking section it is not clear which of the two methods presented for selection of IIS was used (optimization of elastic variables or elastic variables + objective). As growth rate is used as a constraint, this may suggest the former, but not explicitly said and also not clear for researchers starting to explore the subfield.

8.- What does a “a large constant K ” (line 169) means in the context of the MILP problem. How large is it? what was the rationale behind the chosen value? what is the impact of this parameter in the performance of the MILP-based approach?

9.- Table 1 is not informative enough of the modifications, condition parameters and assumptions used to model the conditions of origin for the different proteomic datasets.

10.- Provide statistical tests for the comparison of flux variability ranges across different layers of constraints and add significance arguments to the discussion of these results

Minor comments:

11.- Line 24: replace “Genome-scale model Enzyme constraints” by “Genome-scale model with Enzyme constraints” as originally named by Sánchez, et al., 2017.

12.- Line 65: “Now by enforcing mass conservation over the network, $S_v=0$ ”. The explanation is not as straightforward as mentioned here and has been extensively described in other specialized reviews. Better cite those and rephrase.

11.- Line 76: “an update of an existing open-source enzyme-constraint software: geckopy 3.0” please follow the recommendation in the major concerns and be more descriptive here, the presented geckopy is a software for incorporation of omics constraints into models with enzymatic parameters.

12.- Line 77: “a SBML” acronyms starting with “S” are usually referred to with “an”.

13.- Line 78: recommended to call proteins as “pseudometabolites” in order to avoid confusion.

14.- Line 112: “ v is the vector of reaction variables”. More than 20 years of FBA related papers have named this as a vector of reaction fluxes, recommended to indicate that the prediction outcome is the distribution of fluxes as a separate idea.

15.- Line 116: Provide references to the use of the biomass pseudoreaction or the ATP synthase as objective functions.

16.- Line 154: “a LP problem”, acronyms starting with L are usually referred to with “an”.

17.- Line 159: “the total flux”, not clear if the authors refer to the metabolic flux that those enzymes carry or if they mean total use of elastic variables instead.

18.- There is a gap in between lines 163 and 164 that does not clarify what the listed points are specifically referring to.

19.- Line 225: not clear with what is meant with “when optimizing biomass for all reactions” in this context.

20.- A one line explanation describing the thermodynamic solution will help to clarify this section.

21.- Line 298 and 306: The terms “number of conditions” and “number of samples” is not clear when reading this and also figure 1 with its corresponding caption. What does it mean? Do the authors refer to the identifiers of the conditions/samples (e.g. sample 1, sample 2, ..., sample 17)? Clarifying this will majorly help this section and figure 1.

22.- The average number of relaxed proteins across algorithms reported in lines 299- 302 does not seem to correspond with the counts values shown in figure 1 (y-axis). Or do they refer to different variables?

23.- Line 317: Here fig. 4 is referred to before introducing figure 3, therefore the order of these figures should be interchanged so that it reflects the sequence in the text.

24.- Line 322: typo in “one deviation form this trend”

25.- Current figure 4 shows growth rate sum of relaxation values, but units are not reported for any of the axis. According to equation 8, I assume that the sum of relaxation values is presented in mmol/gDw (such as enzyme usages). It is recommended to convert the sum of relaxation values to mass units, by multiplying the contribution of every flexibilized enzyme by its molecular weight. In this way it is possible to assess what is the proportion of flexibilized data in comparison to the total protein content of the cell (global constraint) in terms of a conserved quantity.

26.- Additionally, I recommend to add a color code or different markers to the data points in figure 4, indicating the modeled conditions, as the association between growth rates and experimental conditions is not provided anywhere else in the text or associated materials. This will help to clarify what do authors mean with expression such as “protein constraints are less important when moved away from optimal conditions” (line 327).

27.- Line 334: the sentence “... samples that exceeded the growth rate of and what were capped ...” is hard to understand, probably missing words between “for” and “and”.

Review comments

Muriel et al. presents a new implementation of *geckopy*, a python version of the GECKO software implemented in MATLAB. The GECKO software is one method for introducing enzyme-constraints in GEMs, and this extension of “standard” GEMs have been proved valuable across a range of organisms and scientific questions. As such, a python implementation that makes this method more accessible will be a good contribution to the COBRA community. Also, by formalizing the enzyme-constraints in SBML language the authors make these models in better compliance with FAIR principles. Furthermore, the authors extend the standard GECKO method to incorporate thermodynamic constraints, an important improvement that further improves the value of the developed *geckopy* 3.0. Finally, the authors use a reduced *E. coli* GEM to evaluate different methods for relaxing experimental constraints that restricts the model from achieving the experimentally observed growth rate. In summary, I find that this work addresses several improvements and questions that in principle could be of broad interest. The software has a decent documentation online and it is easy to install with pip.

Unfortunately, I find that both the software itself (*geckopy* 3.0) and the manuscript are of insufficient comprehensiveness / quality to reach its potential value. First and foremost, I would have liked to see the ability in *geckopy* to actually **make** the ecGEMs, ideally including both the extraction of enzyme coefficients from Brenda (or with DLKcat) and the conversion from a standard GEM to an ecGEM. In this way, *geckopy* would be a true alternative to GECKO in matlab. That said, once you have an ecGEM, *geckopy* 3.0 seems useful for simulating the model with methods like FBA and FVA, and for integration of proteomics data, thermodynamics and metabolomics data. And for t

The manuscript itself seems to be hastily written, it is hard to get the main message and how the results relate to each other and to importance of the developed software. Specific comments are given in bullet points below:

- The authors don't put this work into context, and leaves out relevant work like ECMpy and MOMENT
- The introduction reads strange: first the authors spend several lines on explaining FBA (not sure if this is necessary), then summarizes the paper, line 76-83, and then goes back to the GECKO-formulation.
- In the latest version of GECKO, the stoichiometric coefficient of each enzyme is $Mw/Kcat$, not $1/Kcat$. Why haven't the authors aligned their work with this formulation, which I believe is also used by MOMENT?
- Despite multiple figures, it is not clear what's the recommended method for doing relaxation.
- The formulation of the relaxation linear programs seems to miss the constraint on reaction bounds?
- Also is the numbering of the equations in the text correct, e.g. on line 164 the authors refer to Equation 9 (which is also the one below the bullet point), but should this be Eq. 6?

- The numbering of the equations is messy, without any space between # and the equation.
- Eq. 9, it is not clear that this optimize the original objective, as this seems to minimize Z, while in the original it maximizes Z
- Line 190: two commas next to each other.
- I would like to see the FVA comparison on full GEM (not E. coli core)
 - o Maybe also for yeast to confirm that these trends are more general
- The Enzyme-constrained E.coli GEM has been relaxed in other publications, does the findings here align with previous results? Are the same enzymes relaxed in similar work on yeast or *S. coelicolor*?
- Line 118: Biomass components should sum up to 1 gDW, so the units is actually just 1/h
- Line 197: ODml?
- Line 383: Figure 3, not 3
- Line 384: Write out Glucose, not Glc
- Line 335: rightmost
- I would like to see the jupyter notebooks in the .ipynb format and not .html to be able to reproduce the results.
-

Reviewer #1 (Comments for the Author):

Major concerns:

1.- The geckopy implementation presented here is a software pipeline for integration of omics and thermodynamic constraints into an enzyme-constrained model of metabolism. This is well explained by the title. However, the abstract and introduction sections do not specify that the enzymatic constraints, or catalytic constants are not treated here and are taken as inputs in the geckopy pipeline. The current text may sound self-explanatory to the authors or very specialized researchers, but the readers of Microbiology Spectrum include a broader audience.

We have rephrased a sentence in the abstract to make the requirement of kinetic data clear. The introduction has been expanded to explain in greater detail the rationality and behavior of enzyme-constrained models for a broader audience.

2.- The geckopy implementation presented here is a software pipeline for integration of omics and thermodynamic constraints into an enzyme-constrained model of metabolism. This 2.- The tool is presented as a version 3.0 of the originally proposed geckopy by Sánchez, et al. 2017. At the same time it is highlighted that this is a reconstruction from scratch, which I do agree that is presented here, but breaks strict software versioning practices. A problem, intrinsic to GECKO available at: <https://github.com/SysBioChalmers/GECKO>, is that development of the python module stopped practically at its first version, whilst MATLAB versions presented methodological changes in data integration (GECKO 2.0), and even at the model format level (GECKO 3.0). Therefore, it is strictly recommended to be consistent with the title of this manuscript and refer to the new tool as an independent implementation of data constraints in GECKO throughout the rest of the text and associated materials.

A couple of sentences has been added in the introduction to clarify this. We believe that the version number is perfectly fine with semantic versioning, since a major version number change corresponds to breaking changes and this makes it possible to publish it in PyPi without removing the history of a previously existing package with the same name.

3.- Across the whole manuscript there are several terms that differ from the ones that have been used in the studies of this modeling field. In particular, the term enzyme constraint model is repeatedly used in this manuscript. The habit has been to name these models as enzyme, enzymatically or protein models. Consistency with the terminology of the subfield is recommended unless a re-discussion of the term is presented.

All instances of the concept has been normalized to “enzyme-constrained model” in the text, which is the term used by Sánchez et al., 2017.

4.- The manuscript and supplementary materials do not provide enough detail the parameterization procedure of the reduced metabolic model (i.e. selection of kcat values). It has been reported by several studies that kcat distributions play a major role in flux distributions, therefore, this factor would be expected to majorly impact the conclusions extracted from the results of the flux variability analysis results. It is necessary to provide such parameterization details and complement discussion with more details regarding the selection of parameters. Particularly those discussed at the reaction level. A property of enzyme constrained models is that systems-level distribution of kinetic parameters translates to changes at the reaction level, and even though this is a reduced model, the number of reactions and metabolites suggest that it represents the conjunction of multiple pathways. It is also recommended to mention the sectors of metabolism that this small models accounts for, in order to highlight the relevance of conclusions to a wider public, such as non-modeler microbiologists.

This has been made added to the results of the FVA comparisons. Also, to make it clearer, we have added a note in the results to indicate that the benchmark was performed with eciML1515 and not the reduced model.

5.- The average enzyme saturation across enzymes is analyzed at a genome-scale by using different relaxation algorithms. This analysis returned an overall low saturation value as all individually constrained enzymes were taken into account. Looking separately at the number of “used” enzymes (the flux carrying ones) may offer an additional factor for comparison, and also calculating a separated saturation factor for this group of enzymes may help the discussion of the commonly adopted average saturation factor of 0.5 in models constrained at the protein pool level.

Figure 2 has been modified to show this, agreeing with the reviewer’s note.

6.- It is not clear in the manuscript and methods how the carbon source uptake rate is used as a constraint in addition to those at the proteomic level. This information is of particular importance for constraining a metabolic network, as it is indicative of the metabolic state, mode and even stress levels for a given growth phenotype.

A paragraph in the Methods was updated about the Glucose carbon source and the growth rate.

7.- Comparison of the proposed LP and MILP algorithms for selection of IIS to the brut-force algorithms available in GECKOmat and caffeine is lacking and is strongly recommended, for the sake of providing quantitative evaluation of approaches to the interested community. When referring to the LP approach in the benchmarking section it is not clear which of the two methods presented for selection of IIS was used (optimization of elastic variables or elastic variables + objective). As growth rate is used as a constraint, this may suggest the former, but not explicitly said and also not clear for researchers starting to explore the subfield.

We agree that it is very illustrative and included the results. To make the comparison clearer, the Figure 1 was replaced with a table. The comment about the objective was clarified in the Methods.

8.- What does a “a large constant K” (line 169) means in the context of the MILP problem. How large is it? what was the rationale behind the chosen value? what is the impact of this parameter in the performance of the MILP-based approach?

We added a clarification: “1000, the default upper bound for a COBRA reaction flux, sufficient enough to block the flux through a protein pseudoexchange reaction”.

9.- Table 1 is not informative enough of the modifications, condition parameters and assumptions used to model the conditions of origin for the different proteomic datasets.

A paragraph was added in the methods explaining the conditions themselves in the methods and the reasoning about the modifications to the model.

“About the conditions, they consist on different strains constructed to modulate three substrate limitations: Ammonia limitation (A), Carbon limitation (C) and Ribosome limitation (R). As explained in Table 1, just one modification was done to the model, in the case of ammonia limitation, where the GLUDy reaction was knocked out. The rest of modifications were assumed to be accounted by proteomics data, since they refer to the modulation of the expression of either a protein or the whole proteome (R-limitation). The implementation can be found at S2 Files, *proteomics_data_relaxations.ipynb*.”

10.- Provide statistical tests for the comparison of flux variability ranges across different layers of constraints and add significance arguments to the discussion of these results

We have done as asked, and run pairwise Student’s t-tests and reported them in the following table (nan indicates that they are the same distribution):

	Proteomics	Pool constraint	Thermo	Thermo + Metabolomics	Thermo + Proteomics
FBA	0.0157626	0.792169	0.608079	0.608079	0.0862913
Pool constraint	0.0322534	nan	0.802054	0.802054	nan
Thermo	0.0617078	0.802054	nan	1	0.23612
Thermo + Metabolomics	0.0617078	0.802054	1	nan	0.23612
Thermo + Proteomics	0.498918	0.147727	0.23612	0.23612	nan

The table shows that we cannot reject the null hypothesis that the “Proteomics” method and “Thermo + Proteomics” method fluxes are generated from the same underlying normal distribution under the usual p-value threshold of 0.05. Other comments are that we can reject

the null hypothesis for the “FBA” and “Proteomics” pair or the “Proteomics” and “Pool Constraint”, for instance.

However, we think that this kind of significance test would confuse the readers. First, the assumptions of a t-test do not hold, the width of the fluxes in a linear programming problem are correlated (t-test requires independent and identically distributed observations). Second, the meaning of a p-value in this case is how likely would be to observe a t-statistic at least as extreme as the computed one if we were to generate a new sample of data. However, in this case, there is no such underlying sampling generation process: we are observing the full *population* of variables (not a sample), which would be exactly the same everytime we run the FVA for a particular set of constraints.

Thus, we think that discussing the summary statistics displayed in the box-plot should be enough to discuss the differences and similarities between results.

Minor comments:

11.- Line 24: replace “Genome-scale model Enzyme constraints” by “Genome-scale model with Enzyme constraints” as originally named by Sánchez, et al., 2017.

Fixed.

12.- Line 65: “Now by enforcing mass conservation over the network, $S_v=0$ ”. The explanation is not as straightforward as mentioned here and has been extensively described in other specialized reviews. Better cite those and rephrase.

Added citation and rephrased.

11.- Line 76: “an update of an existing open-source enzyme-constraint software: geckopy 3.0” please follow the recommendation in the major concerns and be more descriptive here, the presented geckopy is a software for incorporation of omics constraints into models with enzymatic parameters.

Rephrased to accommodate the comments.

12.- Line 77: “a SBML” acronyms starting with “S” are usually referred to with “an”.

Fixed.

13.- Line 78: recommended to call proteins as “pseudometabolites” in order to avoid confusion.

Agreed and added.

14.- Line 112: “ v is the vector of reaction variables”. More than 20 years of FBA related papers have named this as a vector of reaction fluxes, recommended to indicate that the prediction outcome is the distribution of fluxes as a separate idea.

Fixed accordingly.

15.- Line 116: Provide references to the use of the biomass pseudoreaction or the ATP synthase as objective functions.

Added the reference.

16.- Line 154: "a LP problem", acronyms starting with L are usually referred to with "an".

Noted.

17.- Line 159: "the total flux", not clear if the authors refer to the metabolic flux that those enzymes carry or if they mean total use of elastic variables instead.

Rephrased to "total value" so that it is less confusing.

18.- There is a gap in between lines 163 and 164 that does not clarify what the listed points are specifically referring to.

That was a formatting error from LaTeX to DOCX, we hope it is clearer now.

19.- Line 225: not clear with what is meant with "when optimizing biomass for all reactions" in this context.

Clarified.

20.- A one line explanation describing the thermodynamic solution will help to clarify this section.

Not sure which section they are referring to.

21.- Line 298: The term the number of conditions is not clear when reading this and also figure 1 with its corresponding caption. What does it mean? Do the authors refer to the identifiers of the conditions/samples (e.g. sample 1, sample 2, ..., sample 17)? Clarifying this will majorly help this section.

Figure one was removed in favor of the table to make it clearer.

22.- The average number of relaxed proteins across algorithms reported in lines 299- 302 does not seem to correspond with the counts values shown in figure 1 (y-axis). Or do they refer to different variables?

Figure 1 was the number of samples (experimental conditions) where a given protein showed up in the IIS. This is different from the average number of proteins in the IIS per method. Anyways, it was removed in favor of the table. Please note that the new table was recomputed and there is some slight numerical variability in the final numbers from run to run.

23.- Line 317: Here fig. 4 is referred to before introducing figure 3, therefore the order of these figures should be interchanged so that it reflects the sequence in the text.

Agreed, we changed it.

24.- Line 322: typo in “one deviation form this trend”

Noted.

25.- Current figure 4 shows growth rate sum of relaxation values, but units are not reported for any of the axis. According to equation 8, I assume that the sum of relaxation values is presented in mmol/gDw (such as enzyme usages). It is recommended to convert the sum of relaxation values to mass units, by multiplying the contribution of every flexibilized enzyme by its molecular weight. In this way it is possible to assess what is the proportion of flexibilized data in comparison to the total protein content of the cell (global constraint) in terms of a conserved quantity.

That is right, we changed it to g/gDw as recommended and annotated the units.

26.- Additionally, I recommend to add a color code or different markers to the data points in figure 4, indicating the modeled conditions, as the association between growth rates and experimental conditions is not provided anywhere else in the text or associated materials. This will help top clarify what do authors mean with expression such as “protein constraints are less important when moved away from optimal conditions” (line 327).

We changed it to “optimal growth conditions” to make it clearer and added colors for the kind of limitation present in the sample.

27.- Line 334: the sentence “... samples that exceeded the growth rate of and what were capped ...” is hard to understand, probably missing words between “for” and “and”.

Added dashes to distinguish it.

Review #2

Unfortunately, I find that both the software itself (geckopy 3.0) and the manuscript are of insufficient comprehensiveness / quality to reach its potential value. First and foremost, I would have liked to see the ability in geckopy to actually make the ecGEMs, ideally including both the extraction of enzyme coefficients from Brenda (or with DLKcat) and the conversion from a standardGEM to an ecGEM. In this way, geckopy would be a true alternative to GECKO in matlab.

The generation of the ecGEM was left out on purpose after discussing it with the authors of GECKOmat to concentrate that part in one place. This aligns with the kind of work that both packages focus on: while GECKOmat has more functionalities about the Kcats (generation pipeline, DLKcat), geckopy is focused on the experimental integration; i.e., the proteomics and metabolomics relaxations. That being said, there has been interest from the GECKO authors to take over geckopy and unify both in a single python package for the next iteration.

That said, once you have an ecGEM, geckopy3.0 seems useful for simulating the models with methods like FBA and FVA, and for integration of proteomics data, thermodynamics and metabolomics data. And for the manuscript itself seems to be hastily written, it is hard to get the main message and how the results relate to each other and to importance of the developed software. Specific comments are given in bullet points below:

- The authors don't put this work into context, and leaves out relevant work like ECMpy and MOMENT.

The introduction was elaborated to expand upon the context.

- The introduction reads strange: first the authors spend several lines on explaining FBA (not sure if this is necessary), then summarizes the paper, line 76-83, and then goes back to the GECKO-formulation.

We think that explaining FBA in the introduction makes sense from the point of view of the broad audience of Microbiology Spectrum that might not be familiar with this field (as pointed out but the other reviewer). We have elaborated further the introduction.

- In the latest version of GECKO, the stoichiometric coefficient of each enzyme is Mw/K_{cat} , not $1/K_{cat}$. Why haven't the authors aligned their work with this formula, which I believe is also used by MOMENT?

At the moment of writing, the latest version of GECKO (3.0) has not been published, so we thought it would be respectful to avoid including it here. Nonetheless, we have implemented support for reading and writing SBML documents with this encoding at the user option (see PR#10), which is not supported by GECKO 3.0 yet, as far as we know.

- Despite multiple figures, it is not clear what's the recommended method for doing relaxation.

We have added a few lines discussing how the LP fashion gives smaller IIS and replaced Figure 1 with a table that better compares the results. We hope that makes it more clear. Another aspect that is already explained is the advantage to explore the relaxation in the LP case. Nonetheless, it is not a black and white answer, as explained in the Methods section:

“The different variants of formulations and objectives reflect different assumptions about the uncertainty of the experimental methods in place. Hence, if it is suspected that the uncertainty is uniformly distributed over all measurements, one of the elastic filtering methods should be chosen over the MILP problem. Correspondingly, if the a priori knowledge points to a reduced subset of the enzymes with high uncertainty, the MILP might be a better fit.”

- The formulation of the relaxation linear programs seems to miss the constraint on reaction bounds?

We added it, thank you for noting it.

- The numbering of the equations is messy, without any space between # and the equation.

We apologize, these are artifacts when converting it from LaTeX to DOCX. We hope that now it is better formatted.

- Eq. 9, it is not clear that this optimize the original objective, as this seems to minimize Z, while in the original it maximizes Z.

We have added a minus to Z to clarify it.

- Line 190: two commas next to each other.

Fixed.

- I would like to see the FVA comparison on full GEM (not E. coli core). Maybe also for yeast to confirm that these trends are more general.

We respectfully disagree. The point is that a naive layering of constraints (and its subsequent relaxation) make not make a more constrained model (thermodynamics + enzyme constraints), which is sufficiently shown for a reduced model that is easier to interpret.

- The Enzyme-constrained E.coli GEM has been relaxed in other publications, does the findings here align with previous results? Are the same enzymes relaxed in similar work on yeast or S. coelicolor?

We have not been able to access the relaxations/flexibilizations performed in Domenzain et al., 2022. The proteome flexibilization that they use is the same greedy relaxation algorithm that is also implemented in geckopy. We have added the results for this algorithm to compare with the rest. We do not know of other publications where this Enzyme-constrained E. coli GEM has been used.

- Line 118: Biomass components should sum up to 1 gDW, so the units is actually just 1/h

Fixed accordingly.

- Line 197: ODml?

Added a dot to clarify it. It uses the same nomenclature as in the original publication (see Supplementary material Figure S2 of Balakrishnan et al., 2022).

- Line 383: Figure 3, not 3

Fixed.

- Line 384: Write out Glucose, not Glc

Fixed.

- Line 335: rightmost

Fixed.

- I would like to see the jupyter notebooks in the .ipynb format and not .html to be able to reproduce the results.

I agree with that and tried it but the submission form does not allow for uploading ipynb extensions. I have added a link to repository that contains the notebooks in jupyter format with the necessary data to run them (referenced in the text). Note that the relaxation notebook was modified to account for the comments of the other reviewer.

September 5, 2023

Dr. Jorge Carrasco Muriel
Danmarks Tekniske Universitet The Novo Nordisk Foundation Center for Biosustainability
Copenhagen
Denmark

Re: Spectrum01705-23R1 (Simultaneous application of enzyme and thermodynamic constraints to metabolic models using an updated Python implementation of GECKO)

Dear Dr. Jorge Carrasco Muriel:

Thank you for submitting your manuscript to Microbiology Spectrum. As you will see your paper is very close to acceptance. Please modify the manuscript along the lines I have recommended. As these revisions are quite minor, I expect that you should be able to turn in the revised paper in less than 30 days, if not sooner. If your manuscript was reviewed, you will find the reviewers' comments below.

When submitting the revised version of your paper, please provide (1) point-by-point responses to the issues raised by the reviewers as file type "Response to Reviewers," not in your cover letter, and (2) a PDF file that indicates the changes from the original submission (by highlighting or underlining the changes) as file type "Marked Up Manuscript - For Review Only". Please use this link to submit your revised manuscript. Detailed instructions on submitting your revised paper are below.

Link Not Available

Sincerely,

Angela Re

Reviewer comments:

Reviewer #2 (Comments for the Author):

I'm glad to see that authors have addressed my comments appropriately. Only two minor comments:

1. There are several spelling errors in line 400-401 in marked-up manuscript (e.g. TCA cylce)
2. I am not sure if it make sense to capitalize metabolite names (Glucose)

Editor minor modifications:

Line 21: "for its enzymes" is not grammatically correct

Lines 45-47 "Additionally, to ensure that enzyme-constrained models follow the community standards, a format for the proteins is postulated." can be move at line 43.

Line 63: $S_v = 0$. It would be useful to introduce what is v here.

Line 79: ... its corresponding reactions ... is not grammatically correct

Line 90: "equal the total amount" is not grammatically correct

Line 128: Figures ??, 1, 3, 2; has to be fixed

Line 129: "to be reproduced at" lacks the verb

Line 139: "pseudorreaction" -> pseudoreaction

Line 149: Flux should be in minor case

Line 195: Why Relaxation is upper case here?

Line 203: $f_{p,i}$ is different from the symbol used in eq. 11

Please check the indices used in eqs. 12 and 13

Line 221: "consist on" -> consist of

Line 251: "which block" is not correct

Line 300: "that reflect the" is not correct

Line 304: "on top th"e is not correct

Line 297: "Enzyme constraint models" is to be fixed

Line 416: "formulation provides" would require a comma in between

Preparing Revision Guidelines

Please return the manuscript within 60 days; if you cannot complete the modification within this time period, please contact me. If you do not wish to modify the manuscript and prefer to submit it to another journal, please notify me of your decision immediately so that the manuscript may be formally withdrawn from consideration by Microbiology Spectrum.

Point-by-point responses: 2nd review

Reviewer #2 (Comments for the Author)

1. There are several spelling errors in line 400-401 in marked-up manuscript (e.g. TCA cylce)

We have fixed it, thank you.

2. I am not sure if it make sense to capitalize metabolite names (Glucose)

We have changed glucose to lower case.

Editor minor modifications

Line 21: "for its enzymes" is not grammarly correct

Fixed, thank you.

Lines 45-47 "Additionally, to ensure that enzyme-constrained models follow the community standards, a format for the proteins is postulated." can be move at line 43.

Instead, we have moved it to line 48, we believe it achieves the desired effect.

Line 63: $S_v = 0$. It would be useful to introduce what is v here.

We have included it now.

Line 90: "equal the total amount" is not grammarly correct

Line 128: Figures ??, 1, 3, 2; has to be fixed

Line 129: "to be reproduced at" lacks the verb

Line 139: "pseudorreaction" -> pseudoreaction

Line 149: Flux should be in minor case

Line 195: Why Relaxation is upper case here?

Fixed.

Line 203: $f_{p,i}$ is diferent from the symbol used in eq. 11

Please check the indices used in eqs. 12 and 13

All indices were changed to "i" on the left hand side for consistency and the equations were updated appropriately.

Line 221: "consist on" -> consist of

Line 251: "which block" is not correct

Line 300: "that reflect the" is not correct

Line 304: "on top th"e is not correct

Line 297: "Enzyme constraint models" is to be fixed

Line 416: "formulation provides" would require a comma in between

All of the above were corrected accordingly.

Point-by-point responses: 1st review

Reviewer #1 (Comments for the Author):

Major concerns:

1.- The geckopy implementation presented here is a software pipeline for integration of omics and thermodynamic constraints into an enzyme-constrained model of metabolism. This is well explained by the title. However, the abstract and introduction sections do not specify that the enzymatic constraints, or catalytic constants are not treated here and are taken as inputs in the geckopy pipeline. The current text may sound self-explanatory to the authors or very specialized researchers, but the readers of Microbiology Spectrum include a broader audience.

We have rephrased a sentence in the abstract to make the requirement of kinetic data clear. The introduction has been expanded to explain in greater detail the rationality and behavior of enzyme-constrained models for a broader audience.

2.- The geckopy implementation presented here is a software pipeline for integration of omics and thermodynamic constraints into an enzyme-constrained model of metabolism. This 2.- The tool is presented as a version 3.0 of the originally proposed geckopy by Sánchez, et al. 2017. At the same time it is highlighted that this is a reconstruction from scratch, which implies independent development, which I do agree that is presented here, but breaks strict software versioning practices. A problem, intrinsic to GECKO available at: <https://github.com/SysBioChalmers/GECKO>, is that development of the python module stopped practically at its first version, whilst MATLAB versions presented methodological changes in data integration (GECKO 2.0), and even at the model format level (GECKO 3.0). Therefore, it is strictly recommended to be consistent with the title of this manuscript and refer to the new tool as an independent implementation of data constraints in GECKO throughout the rest of the text and associated materials.

A couple of sentences has been added in the introduction to clarify this. We believe that the version number is perfectly fine with semantic versioning, since a major version number change

corresponds to breaking changes and this makes it possible to publish it in PyPi without removing the history of a previously existing package with the same name.

3.- Across the whole manuscript there are several terms that differ from the ones that have been used in the studies of this modeling field. In particular, the term enzyme constraint model is repeatedly used in this manuscript. The habit has been to name these models as enzyme, enzymatically or protein models. Consistency with the terminology of the subfield is recommended unless a re-discussion of the term is presented.

All instances of the concept has been normalized to “enzyme-constrained model” in the text, which is the term used by Sánchez et al., 2017.

4.- The manuscript and supplementary materials do not provide enough detail the parameterization procedure of the reduced metabolic model (i.e. selection of kcat values). It has been reported by several studies that kcat distributions play a major role in flux distributions, therefore, this factor would be expected to majorly impact the conclusions extracted from the results of the flux variability analysis results. It is necessary to provide such parameterization details and complement discussion with more details regarding the selection of parameters. Particularly those discussed at the reaction level. A property of enzyme constrained models is that systems-level distribution of kinetic parameters translates to changes at the reaction level, and even though this is a reduced model, the number of reactions and metabolites suggest that it represents the conjunction of multiple pathways. It is also recommended to mention the sectors of metabolism that this small models accounts for, in order to highlight the relevance of conclusions to a wider public, such as non-modeler microbiologists.

This has been made added to the results of the FVA comparisons. Also, to make it clearer, we have added a note in the results to indicate that the benchmark was performed with eciML1515 and not the reduced model.

5.- The average enzyme saturation across enzymes is analyzed at a genome-scale by using different relaxation algorithms. This analysis returned an overall low saturation value as all individually constrained enzymes were taken into account. Looking separately at the number of “used” enzymes (the flux carrying ones) may offer an additional factor for comparison, and also calculating a separated saturation factor for this group of enzymes may help the discussion of the commonly adopted average saturation factor of 0.5 in models constrained at the protein pool level.

Figure 2 has been modified to show this, agreeing with the reviewer’s note.

6.- It is not clear in the manuscript and methods how the carbon source uptake rate is used as a constraint in addition to those at the proteomic level. This information is of particular importance for constraining a metabolic network, as it is indicative of the metabolic state, mode and even stress levels for a given growth phenotype.

A paragraph in the Methods was updated about the Glucose carbon source and the growth rate.

7.- Comparison of the proposed LP and MILP algorithms for selection of IIS to the brut-force algorithms available in GECKOmat and caffeine is lacking and is strongly recommended, for the sake of providing quantitative evaluation of approaches to the interested community. When referring to the LP approach in the benchmarking section it is not clear which of the two methods presented for selection of IIS was used (optimization of elastic variables or elastic variables + objective). As growth rate is used as a constraint, this may suggest the former, but not explicitly said and also not clear for researchers starting to explore the subfield.

We agree that it is very illustrative and included the results. To make the comparison clearer, the Figure 1 was replaced with a table. The comment about the objective was clarified in the Methods.

8.- What does a “a large constant K” (line 169) means in the context of the MILP problem. How large is it? what was the rationale behind the chosen value? what is the impact of this parameter in the performance of the MILP-based approach?

We added a clarification: “1000, the default upper bound for a COBRA reaction flux, sufficient enough to block the flux through a protein pseudoexchange reaction”.

9.- Table 1 is not informative enough of the modifications, condition parameters and assumptions used to model the conditions of origin for the different proteomic datasets.

A paragraph was added in the methods explaining the conditions themselves in the methods and the reasoning about the modifications to the model.

“About the conditions, they consist on different strains constructed to modulate three substrate limitations: Ammonia limitation (A), Carbon limitation (C) and Ribosome limitation (R). As explained in Table 1, just one modification was done to the model, in the case of ammonia limitation, where the GLUDy reaction was knocked out. The rest of modifications were assumed to be accounted by proteomics data, since they refer to the modulation of the expression of either a protein or the whole proteome (R-limitation). The implementation can be found at S2 Files, *proteomics_data_relaxations.ipynb*.”

10.- Provide statistical tests for the com on full GEM (not E. coli core). Maybe also for yeast to confirm that these trends are more general.

We respectfully disagree. The point is that a naive layering of constraints (and its subsequent relaxation) make not make a more constrained model (thermodynamics + enzyme constraints), which is sufficiently shown for a reduced model that is easier to interpret.

- The Enzyme-constrained E.coli GEM has been relaxed in other publications, does the findings here align with previous results? Are the same enzymes relaxed in similar work on yeast or S. coelicolor?

We have not been able to access the relaxations/flexibilizations performed in Domenzain et al., 2022. The proteome flexiblization that they use is the same greedy relaxation algorithm that is

also implemented in geckopy. We have added the results for this algorithm to compare with the rest. We do not know of other publications where this Enzyme-constrained E. coli GEM has been used.

- Line 118: Biomass components should sum up to 1 gDW, so the units is actually just 1/h

Fixed accordingly.

- Line 197: ODml?

Added a dot to clarify it. It uses the same nomenclature as in the original publication (see Supplementary material Figure S2 of Balakrishnan et al., 2022).

- Line 383: Figure 3, not 3

Fixed.

- Line 384: Write out Glucose, not Glc

Fixed.

- Line 335: rightmost

Fixed.

- I would like to see the jupyter notebooks in the .ipynb format and not .html to be able to reproduce the results.

I agree with that and tried it but the submission form does not allow for uploading ipynb extensions. I have added a link to repository that contains the notebooks in jupyter format with the necessary data to run them (referenced in the text). Note that the relaxation notebook was modified to account for the comments of the other reviewer. parison of flux variability ranges across different layers of constraints and add significance arguments to the discussion of these results

We have done as asked, and run pairwise Student's t-tests and reported them in the following table (nan indicates that they are the same distribution):

	Proteomics	Pool constraint	Thermo	Thermo + Metabolomics	Thermo + Proteomics
FBA	0.0157626	0.792169	0.608079	0.608079	0.0862913
Pool constraint	0.0322534	nan	0.802054	0.802054	nan
Thermo	0.0617078	0.802054	nan	1	0.23612
Thermo +	0.061707	0.802054	1	nan	0.23612

	Proteomics	Pool constraint	Thermo	Thermo + Metabolomics	Thermo + Proteomics
Metabolomics	8				
Thermo + Proteomics	0.498918	0.147727	0.23612	0.23612	nan

The table shows that we cannot reject the null hypothesis that the “Proteomics” method and “Thermo + Proteomics” method fluxes are generated from the same underlying normal distribution under the usual p-value threshold of 0.05. Other comments are that we can reject the null hypothesis for the “FBA” and “Proteomics” pair or the “Proteomics” and “Pool Constraint”, for instance.

However, we think that this kind of significance test would confuse the readers. First, the assumptions of a t-test do not hold, the width of the fluxes in a linear programming problem are correlated (t-test requires independent and identically distributed observations). Second, the meaning of a p-value in this case is how likely would be to observe a t-statistic at least as extreme as the computed one if we were to generate a new sample of data. However, in this case, there is no such underlying sampling generation process: we are observing the full *population* of variables (not a sample), which would be exactly the same everytime we run the FVA for a particular set of constraints.

Thus, we think that discussing the summary statistics displayed in the box-plot should be enough to discuss the differences and similarities between results.

Minor comments:

11.- Line 24: replace “Genome-scale model Enzyme constraints” by “Genome-scale model with Enzyme constraints” as originally named by Sánchez, et al., 2017.

Fixed.

12.- Line 65: “Now by enforcing mass conservation over the network, $S_v=0$ ”. The explanation is not as straightforward as mentioned here and has been extensively described in other specialized reviews. Better cite those and rephrase.

Added citation and rephrased.

11.- Line 76: “an update of an existing open-source enzyme-constraint software: geckopy 3.0” please follow the recommendation in the major concerns and be more descriptive here, the presented geckopy is a software for incorporation of omics constraints into models with enzymatic parameters.

Rephrased to accommodate the comments.

12.- Line 77: “a SBML” acronyms starting with “S” are usually referred to with “an”.

Fixed.

13.- Line 78: recommended to call proteins as “pseudometabolites” in order to avoid confusion.

Agreed and added.

14.- Line 112: “ v is the vector of reaction variables”. More than 20 years of FBA related papers have named this as a vector of reaction fluxes, recommended to indicate that the prediction outcome is the distribution of fluxes as a separate idea.

Fixed accordingly.

15.- Line 116: Provide references to the use of the biomass pseudoreaction or the ATP synthase as objective functions.

Added the reference.

16.- Line 154: “a LP problem”, acronyms starting with L are usually referred to with “an”.

Noted.

17.- Line 159: “the total flux”, not clear if the authors refer to the metabolic flux that those enzymes carry or if they mean total use of elastic variables instead.

Rephrased to “total value” so that it is less confusing.

18.- There is a gap in between lines 163 and 164 that does not clarify what the listed points are specifically referring to.

That was a formatting error from LaTeX to DOCX, we hope it is clearer now.

19.- Line 225: not clear with what is meant with “when optimizing biomass for all reactions” in this context.

Clarified.

20.- A one line explanation describing the thermodynamic solution will help to clarify this section.

Not sure which section they are referring to.

21.- Line 298: The term the number of conditions is not clear when reading this and also figure 1 with its corresponding caption. What does it mean? Do the authors refer to the identifiers of the conditions/samples (e.g. sample 1, sample 2, ..., sample 17)? Clarifying this will majorly help this section.

Figure one was removed in favor of the table to make it clearer.

22.- *The average number of relaxed proteins across algorithms reported in lines 299- 302 does not seem to correspond with the counts values shown in figure 1 (y-axis). Or do they refer to different variables?*

Figure 1 was the number of samples (experimental conditions) where a given protein showed up in the IIS. This is different from the average number of proteins in the IIS per method. Anyways, it was removed in favor of the table. Please note that the new table was recomputed and there is some slight numerical variability in the final numbers from run to run.

23.- *Line 317: Here fig. 4 is referred to before introducing figure 3, therefore the order of these figures should be interchanged so that it reflects the sequence in the text.*

Agreed, we changed it.

24.- *Line 322: typo in “one deviation form this trend”*

Noted.

25.- *Current figure 4 shows growth rate sum of relaxation values, but units are not reported for any of the axis. According to equation 8, I assume that the sum of relaxation values is presented in mmol/gDw (such as enzyme usages). It is recommended to convert the sum of relaxation values to mass units, by multiplying the contribution of every flexibilized enzyme by its molecular weight. In this way it is possible to assess what is the proportion of flexibilized data in comparison to the total protein content of the cell (global constraint) in terms of a conserved quantity.*

That is right, we changed it to g/gDw as recommended and annotated the units.

26.- *Additionally, I recommend to add a color code or different markers to the data points in figure 4, indicating the modeled conditions, as the association between growth rates and experimental conditions is not provided anywhere else in the text or associated materials. This will help top clarify what do authors mean with expression such as “protein constraints are less important when moved away from optimal conditions” (line 327).*

We changed it to “optimal growth conditions” to make it clearer and added colors for the kind of limitation present in the sample.

27.- *Line 334: the sentence “... samples that exceeded the growth rate of and what were capped ...” is hard to understand, probably missing words between “for” and “and”.*

Added dashes to distinguish it.

Review #2

Unfortunately, I find that both the software itself (geckopy 3.0) and the manuscript are of insufficient comprehensiveness / quality to reach its potential value. First and foremost, I would have liked to see the ability in geckopy to actually make the ecGEMs, ideally including both the extraction of enzyme coefficients from Brenda (or with DLKcat) and the conversion from a standardGEM to an ecGEM. In this way, geckopy would be a true alternative to GECKO in matlab.

The generation of the ecGEM was left out on purpose after discussing it with the authors of GECKOmat to concentrate that part in one place. This aligns with the kind of work that both packages focus on: while GECKOmat has more functionalities about the Kcats (generation pipeline, DLKcat), geckopy is focused on the experimental integration; i.e., the proteomics and metabolomics relaxations. That being said, there has been interest from the GECKO authors to take over geckopy and unify both in a single python package for the next iteration.

That said, once you have an ecGEM, geckopy3.0 seems useful for simulating the models with methods like FBA and FVA, and for integration of proteomics data, thermodynamics and metabolomics data. And for the manuscript itself seems to be hastily written, it is hard to get the main message and how the results relate to each other and to importance of the developed software. Specific comments are given in bullet points below:

- The authors don't put this work into context, and leaves out relevant work like ECMpy and MOMENT.

The introduction was elaborated to expand upon the context.

- The introduction reads strange: first the authors spend several lines on explaining FBA (not sure if this is necessary), then summarizes the paper, line 76-83, and then goes back to the GECKO-formulation.

We think that explaining FBA in the introduction makes sense from the point of view of the broad audience of Microbiology Spectrum that might not be familiar with this field (as pointed out but the other reviewer). We have elaborated further the introduction.

- In the latest version of GECKO, the stoichiometric coefficient of each enzyme is $Mw/Kcat$, not $1/Kcat$. Why haven't the authors aligned their work with this formula, which I believe is also used by MOMENT?

At the moment of writing, the latest version of GECKO (3.0) has not been published, so we thought it would be respectful to avoid including it here. Nonetheless, we have implemented support for reading and writing SBML documents with this encoding at the user option (see PR#10), which is not supported by GECKO 3.0 yet, as far as we know.

- Despite multiple figures, it is not clear what's the recommended method for doing relaxation.

We have added a few lines discussing how the LP fashion gives smaller IIS and replaced Figure 1 with a table that better compares the results. We hope that makes it more clear. Another

aspect that is already explained is the advantage to explore the relaxation in the LP case. Nonetheless, it is not a black and white answer, as explained in the Methods section:

“The different variants of formulations and objectives reflect different assumptions about the uncertainty of the experimental methods in place. Hence, if it is suspected that the uncertainty is uniformly distributed over all measurements, one of the elastic filtering methods should be chosen over the MILP problem. Correspondingly, if the a priori knowledge points to a reduced subset of the enzymes with high uncertainty, the MILP might be a better fit.”

- The formulation of the relaxation linear programs seems to miss the constraint on reaction bounds?

We added it, thank you for noting it.

- The numbering of the equations is messy, without any space between # and the equation.

We apologize, these are artifacts when converting it from LaTeX to DOCX. We hope that now it is better formatted.

- Eq. 9, it is not clear that this optimize the original objective, as this seems to minimize Z, while in the original it maximizes Z.

We have added a minus to Z to clarify it.

- Line 190: two commas next to each other.

Fixed.

- I would like to see the FVA comparison on full GEM (not E. coli core). Maybe also for yeast to confirm that these trends are more general.

We respectfully disagree. The point is that a naive layering of constraints (and its subsequent relaxation) make not make a more constrained model (thermodynamics + enzyme constraints), which is sufficiently shown for a reduced model that is easier to interpret.

- The Enzyme-constrained E.coli GEM has been relaxed in other publications, does the findings here align with previous results? Are the same enzymes relaxed in similar work on yeast or S. coelicolor?

We have not been able to access the relaxations/flexibilizations performed in Domenzain et al., 2022. The proteome flexibilization that they use is the same greedy relaxation algorithm that is also implemented in geckopy. We have added the results for this algorithm to compare with the rest. We do not know of other publications where this Enzyme-constrained E. coli GEM has been used.

- Line 118: Biomass components should sum up to 1 gDW, so the units is actually just 1/h

Fixed accordingly.

- *Line 197: ODml?*

Added a dot to clarify it. It uses the same nomenclature as in the original publication (see Supplementary material Figure S2 of Balakrishnan et al., 2022).

- *Line 383: Figure 3, not 3*

Fixed.

- *Line 384: Write out Glucose, not Glc*

Fixed.

- *Line 335: rightmost*

Fixed.

- *I would like to see the jupyter notebooks in the .ipynb format and not .html to be able to reproduce the results.*

I agree with that and tried it but the submission form does not allow for uploading ipynb extensions. I have added a link to repository that contains the notebooks in jupyter format with the necessary data to run them (referenced in the text). Note that the relaxation notebook was modified to account for the comments of the other reviewer.

September 11, 2023

Dr. Jorge Carrasco Muriel
Danmarks Tekniske Universitet The Novo Nordisk Foundation Center for Biosustainability
Copenhagen
Denmark

Re: Spectrum01705-23R2 (Simultaneous application of enzyme and thermodynamic constraints to metabolic models using an updated Python implementation of GECKO)

Dear Dr. Jorge Carrasco Muriel:

Your manuscript has been accepted, and I am forwarding it to the ASM Journals Department for publication. You will be notified when your proofs are ready to be viewed.

Sincerely,

Angela Re
Editor, Microbiology Spectrum
